# Structural insights into proprotein convertase activation facilitate the engineering of highly specific furin inhibitors

Rupert Klaushofer [1,2,4], Konstantin Bloch[3,4], Luisa Susanna Eder [1,2], Simone Marzaro[1], Mario Schubert [1], Eva Böttcher-Friebertshäuser[3], Hans Brandstetter[1,2] & Sven O. Dahms [1,2] ✉

Proprotein convertases (PCs), including furin and PC1/3 among nine mammalian homologues, mediate the maturation of numerous secreted substrates by proteolytic cleavage. Disbalance of PC activity is associated with diseases like cancer, fibrosis, neurodegeneration and infections. Therefore, PCs are promising drug targets for the treatment of many diseases. However, the highly conserved active site of PCs complicates the development of specific inhibitors. Here we investigate the activation mechanism of PCs using X-ray crystallography and biochemical methods. The structure-based optimization of the multibasic secondary cleavage site loop not only prevents the prodomain's proteolytic cleavage but also increases its inhibition of furin. Combination of cleavage-resistant PC1/3-prodomain variants and a furin-specific nanobody in fusion proteins reveal very potent inhibitors ($K_i = 1.2$ pM) with a more than 25,000-fold higher specificity for furin compared to the closely related PC-member PCSK5. Such fusion proteins effectively suppress the replication of a furin-dependent H7N7-influenza virus in cell-based assays.

Proprotein convertases (PCs) are $Ca^{2+}$-dependent serine endoproteinases harboring a catalytic domain with structural homology to subtilisin[1]. The so called kexin/furin-like mammalian PC family members (furin, PC1/3, PC2, PC4, PACE4, PC5/6 and PC7) recognize multibasic substrate sequences and cleave after the common pattern (R/K) Xn(R)↓ (where $n = 0, 2, 4$, or 6; X represents any amino acid and "↓" marks the scissile peptide bond)[2,3]. Furin, often regarded as the prototypical PC, is the best-characterized member of this protease family and prefers the consensus cleavage motif R-X-K/R-R↓[4,5]. The zymogens of the furin-like PCs share a multidomain architecture comprising an N-terminal pro-domain, a subtilisin-like catalytic domain, and a P-domain (also called HomoB-domain)[1]. These domains are essentially required for maturation of the zymogen and proteolytic activity.

PCs trigger the maturation of a broad range of proteins in the secretory pathway by post-translational proteolytic modification[1–3].

Physiological substrates include the precursors of growth factors, hormones, receptors, matrix metalloproteases, and coagulation factors. Disbalance of PC activity and over- or under-activation of the substrates is associated with many diseases. For instance, unbalanced furin activity promotes cancer-related processes like cell proliferation, metastasis or vascularization[6]. Thus, furin is a potential target for cancer therapy[7,8]. In addition to endogenous substrates, PCs processes proteins of many pathogens[2,3]. Proprotein convertase inhibitors effectively suppressed the maturation of bacterial toxins[9] and viral envelop proteins, including influenza virus hemagglutinin[10,11] and the spike protein of severe acute respiratory syndrome coronavirus 2 (SARS-CoV-2)[12]. Consequently, PC-inhibitors are promising therapeutics with a broad range of potential applications. To avoid potential side effects in pharmacological applications, inhibitors should be as specific as possible and preferentially inhibit only the

[1]Department of Biosciences and Medical Biology, University of Salzburg, Salzburg, Austria. [2]Center for Tumor Biology and Immunology (CTBI), University of Salzburg, Salzburg, Austria. [3]Institute of Virology, Philipps University, Marburg, Germany. [4]These authors contributed equally: Rupert Klaushofer, Konstantin Bloch. ✉e-mail: sven.dahms@plus.ac.at

**Table 1 | $K_i$-values of PC1/3-prodomain secondary cleavage site mutants M1-5 towards furin**

```
        28              77 78 80 81           110
  H-GKRQ...FKHKNHP RRSRR SAF...RSKR-OH
              Secondary cleavage site motive
```

| No. | 77 | 78 | 80 | 81 | $K_i{}^a$ (nM) |
|-----|-----|-----|-----|-----|-----|
| M1 | R | A | R | R | $0.61 \pm 0.02$ |
| M2 | A | R | A | A | $0.32 \pm 0.03$ |
| M3 | A | A | A | A | $1.50 \pm 0.02$ |
| M4 | R | A | A | R | $0.33 \pm 0.03$ |
| M5 | R | K | A | R | $0.042 \pm 0.002$ |

$^a$pH 7.4, 150 mM NaCl, 37 °C.

targeted PC-family member[7,8]. The development of selective inhibitors, however, is complicated by the high conservation of the PCs' active site clefts[13,14]. Targeting less conserved surface patches might be a promising strategy to overcome this limitation. For instance, small molecule inhibitors targeting a cryptic binding pocket of the PCs showed a higher selectivity towards furin than canonical substrate-like inhibitors[15–17]. Inhibitory nanobodies directed against the less conserved P-domain revealed a superior selectivity for furin[18,19]. However, their potency was lower than active site-directed, peptidic, and non-peptidic inhibitors[20].

The prodomains of the PCs play a central role as intramolecular chaperones and as regulators of the auto-activation mechanisms of these proteases[1]. Activation of furin is triggered by two sequential autocatalytic cleavages and subsequent release of the prodomain. The first auto-proteolysis event occurs in the linker region between the prodomain and the catalytic domain directly after folding of the protease in the endoplasmic reticulum[21,22]. Trafficking of furin to the late Golgi is accompanied by a drop in pH to 6.0, triggering a second cleavage within the prodomain[23]. A pH-sensing function was reported for the PC-prodomains[24] that involves conserved histidine residues[25–27]. A change of the conformation at acidic pH probably results in a higher cleavage susceptibility of the loop that carries the secondary cleavage site (referred to as "secondary cleavage site loop"). This mechanism is likely conserved in all furin-like PCs. Although the prodomains of PC4 and PC7[28,29] do not contain a canonical secondary cleavage site, pH-dependent structural changes are thought to play a role for the activation of these proteases.

The PC-prodomains provide an interesting scaffold for inhibitor development. They can be produced in high amounts in bacteria, and potent competitive inhibition of PCs has been demonstrated (up to sub-nM $K_i$)[30,31]. However, wild-type PC-prodomains come with several major drawbacks: (1) they are rapidly degraded by active PCs due to the auto-activation mechanism, (2) they show a limited specificity and typically inhibit several PC family members, (3) they are less stable in acidic compartments of the secretory system where many substrates are processed by the PCs.

In this work, we investigate the structural and functional basis of PC-activation to engineer prodomain-based PC-inhibitors. Combining stabilized PC1/3-prodomain variants with a furin-specific nanobody, we present a highly furin-specific inhibitory fusion protein with potent activity against influenza A virus.

## Results

### Engineering the PC1/3-prodomain for stable and potent furin inhibition

Due to the secondary cleavage site, the wild-type PC-prodomains are both substrates and inhibitors. This equilibrium is influenced by pH, reflecting the auto-activation process of the PCs. Typically, a shift to

lower pH (~6.0) triggers secondary cleavage and thus activation of the PCs. This process can be mimicked mixing the PC1/3-prodomain and furin in a limited proteolysis setup (Fig. S1A). Indeed, we observed degradation of the PC1/3-prodomain, which was increased at pH 6.0 compared to pH 7.4. The PC1/3-prodomain was stable in the absence of furin.

To engineer stable furin inhibitors based on the PC1/3-prodomain, we produced several proteolysis-resistant variants (Table 1). The secondary cleavage site loop of the PC1/3-prodomain contains the multi-basic motif (77)R-S-R-R, including 4 potential cleavage sites: (77)R-R-S-R-R↓, (77)R-R-S-R↓, (77)R-R↓ and (80)R-R↓. As reported by Rabah et al., furin cleavage was abolished for the R78A-mutant (**M1**, Fig. S1B[31],). **M1** inhibited furin with a $K_i$ of $0.61 \pm 0.02$ nM, which indicates the formation of a stable furin:prodomain-complex (Table 1 and Fig. S2A).

The complementary triple mutant (77)A-R-S-A-A (**M2**) was also resistant to proteolysis by furin (Fig. S1C) but revealed more potent inhibition compared to **M1** ($K_i = 0.32 \pm 0.03$ nM, Table 1 and Fig. S2B). The cleavage-resistant mutant (77)A-A-S-A-A (**M3**) (Fig. S1D) showed weaker inhibition than **M1** ($K_i = 1.50 \pm 0.02$ nM, Table 1 and Fig. S2C). These results indicate a role of the secondary cleavage site loop for prodomain:protease interactions with a strong contribution of Arg78. Intrigued by this finding, we tested the mutant (77)R-A-S-A-R (**M4**), which revealed a $K_i$ of $0.33 \pm 0.03$ nM (Fig. S1E, Table 1, and Fig. S2D). Apparently, the presence of Arg77 and Arg81 can compensate for the lack of Arg78. Comparing **M1** and **M4**, absence of Arg80 even improved the affinity of the prodomain for furin. We created the mutant (77)R-K-S-A-R (**M5**) that combined these features, carrying positive charges only at positions 77,78, and 81, as well as eliminating all possible multi-basic cleavage motives (Lysine is not recognized by furin at the P4 position). In excellent agreement with the findings above, **M5** ($K_i = 0.042 \pm 0.002$ nM, Table 1 and Fig. S2E) was ~8-fold more potent than **M2** and ~36-fold more potent than **M3** as well as resistant to auto-activation (Fig. S1F).

### The secondary cleavage site motif influences the structural stability of the prodomain and its pH-sensor

To test the influence of the secondary cleavage site motif on the overall structural integrity of the PC1/3-prodomain we measured the melting temperatures ($T_m$) using nano-DSF at pH 6.0 (Fig. S3A) and pH 7.4 (Fig. S3B). For the wild-type protein, we measured $T_m$-values of $40.9 \pm 0.1$ °C and $48.2 \pm 0.0$ °C at pH 6.0 and pH 7.4, respectively (Table 2). This indicates a reduction of the structural stability of the PC1/3-prodomain at pH 6.0. Similar pH-dependent differences of the $T_m$-values were observed for **M2** and **M5** (Table 2) whereby these mutants showed a slight increase of stability independent of pH. For **M5**, $T_m$-values of $41.2 \pm 0.0$ °C and $48.9 \pm 0.1$ °C were observed at pH 6.0 and pH 7.4, respectively (Table 2). All Arg78Ala mutants, however, showed a structural destabilization at pH 7.4 but not at pH 6.0. For **M1** we measured $T_m$-values of $41.1 \pm 0.2$ °C and $43.7 \pm 0.0$ °C at pH 6.0 and pH 7.4, respectively. This is a surprising result because a pH-sensing function has been reported only for specific histidine residues of the PC1/3-prodomain so far. In conclusion, the basic residue at position 78 is apparently not only required for the affinity to furin but also to maintain the structural stability of the PC1/3-prodomain. For these reasons, we used the mutants **M2** and **M5** as basis for further experiments.

### X-ray structures in complex with furin at pH 5.9 and pH 7.0 reveal the pH-sensing mechanism of the PC1/3-prodomain

The pH-dependent destabilization is an unwanted feature of the PC1/3-prodomain for an application as inhibitory protein. To investigate its pH-sensing mechanism, we aimed especially on crystallization of the PC1/3-prodomain:furin complex at the relevant pH. Indeed, we could obtain crystals of **M5**:furin and **M2**:furin at pH 5.9 and pH 7.0,

**Table 2 | pH-dependent melting temperatures and $K_i$-values of the PC1/3-prodomain mutants M1-10 compared to the wild-type (WT) protein**

```
                    28              72       77 78  80 81              110
                    |               |        |  |   |  |               |
         H-GKRQ...FKHKNHPRRSRRSAF...RSKR-OH
                         Secondary cleavage site motive
```

| No. | 72 | 77 | 78 | 80 | 81 | pH 7.4[a] $K_i$ (nM) | pH 7.4[a] $T_m$ (°C) Furin[b] - | pH 7.4[a] $T_m$ (°C) Furin[b] + | pH 6.0[a] $K_i$ (nM) | pH 6.0[a] $T_m$ (°C) Furin[b] - | pH 6.0[a] $T_m$ (°C) Furin[b] + |
|---|---|---|---|---|---|---|---|---|---|---|---|
| WT | H | R | R | R | R | | 48.2 ± 0.0 | | | 40.9 ± 0.1 | |
| M1 | H | R | A | R | R | 0.56 ± 0.03 | 43.7 ± 0.0 | 68.8 ± 0.1 | 2.2 ± 0.1 | 41.1 ± 0.2 | 67.0 ± 0.0 |
| M2 | H | A | R | A | A | 0.73 ± 0.02 | 50.1 ± 0.0 | 71.2 ± 0.1 | 0.88 ± 0.03 | 41.9 ± 0.1 | 68.7 ± 0.1 |
| M3 | H | A | A | A | A | | 44.5 ± 0.0 | 68.5 ± 0.0 | | 41.1 ± 0.0 | 67.8 ± 0.0 |
| M4 | H | R | A | A | R | | 44.7 ± 0.1 | 69.6 ± 0.1 | | 41.0 ± 0.1 | 67.8 ± 0.0 |
| M5 | H | R | K | A | R | 0.139 ± 0.008 | 48.9 ± 0.1 | 75.1 ± 0.1 | 0.35 ± 0.02 | 41.2 ± 0.0 | 70.4 ± 0.1 |
| M6 | N | R | R | R | R | | 50.0 ± 0.0 | | | 42.0 ± 0.1 | |
| M7 | D | R | R | R | R | | 48.2 ± 0.1 | | | 41.3 ± 0.0 | |
| M8 | L | R | R | R | R | | 52.5 ± 0.2 | | | 45.4 ± 0.1 | |
| M9 | L | A | R | A | A | 0.62 ± 0.03 | 54.3 ± 0.0 | 72.5 ± 0.0 | 0.35 ± 0.01 | 47.7 ± 0.1 | 71.8 ± 0.0 |
| M10 | L | R | K | A | R | 0.101 ± 0.007 | 53.0 ± 0.0 | 75.5 ± 0.1 | 0.14 ± 0.01 | 46.3 ± 0.0 | 73.4 ± 0.1 |

[a]300 mM NaCl, 25 °C.
[b]$T_m$ of isolated furin: 65.6 ± 0.0 °C (pH 7.4) and 63.1 ± 0.1 °C (pH 6.0).

respectively. These crystals were too small to obtain complete datasets of good quality from single crystals. Thus, we applied a microcrystallography approach for data collection and measured partial datasets from many small crystals that were merged afterwards to obtain a complete dataset. The structures of **M2**:furin and **M5**:furin were refined to 2.0 Å and 2.4 Å resolution (Table S1), respectively. Four and three copies of the PC1/3-prodomain:furin complexes were observed in the asymmetric unit for **M2**:furin and **M5**:furin, respectively.

In both complex structures, the globular part of the prodomain is bound to the western rim of the substrate binding pocket (Fig. 1 and Fig. S4). The C-terminal residues 104-110 extend over all non-primed substrate binding pockets of furin (S1-S6). For **M2**:furin and **M5**:furin analysis with PISA revealed average interaction interface areas of 1216 ± 10 Å$^2$ and 1228 ± 7 Å$^2$, respectively. This indicates a very similar overall binding topology of the PC1/3-prodomain at both pH values (Fig. 1B). This is also reflected by high structural similarities between the **M2**:furin and **M5**:furin complexes, indicated by the average RMSD value of 0.200 ± 0.013 Å. The globular part of the prodomain (without the C-terminal residues, Arg29-Glu104) of **M2**:furin and of **M5**:furin covers average interaction interface areas of 644 ± 5 Å$^2$ and of 675 ± 10 Å$^2$, respectively. This interaction is largely dominated by hydrophobic interactions. The only directed interactions found between the globular part of the prodomain and furin are two hydrogen bonds between the side chain of Ser63 of the prodomain and the carbonyl oxygen of Gly241 of furin, as well as between the side chain of Gln102 of the prodomain and the carbonyl oxygen of Gly241 of furin (Fig. S5). The P1 (Arg110)-, P2 (Lys109)- and P4 (Arg107)-residues of the prodomain's extended C-terminus mediate very similar interactions with the substrate binding cleft as reported for substrate-like inhibitors (Fig. 1A)[5,32]. The amide-nitrogen of the P6-residue (Lys105) forms an additional hydrogen bond with the carbonyl-oxygen of Val231 at the substrate binding cleft of furin (Fig. S5).

The main difference between **M2**:furin and **M5**:furin is found for the secondary cleavage site loop between Lys73 and Ala83. This region was well structured in **M2**:furin at pH 7.0, but no electron density was observed in **M5**:furin at pH 5.9. This region was not involved in crystal contacts of the **M2**:furin complex. We also applied crystalline material of the **M5**:furin complex on a gel, showing that the prodomain was intact (Fig. S6). Apparently, the secondary cleavage site loop adopts a specific conformation at pH 7.0 but gets unstructured at pH 5.9. At pH 7.0 the peptide bonds between Ala77 and Ala80 (Arg77 and Arg80 of the wild-type) were found in a beta-strand-like conformation Fig. 2A. The carbonyl oxygen and amide nitrogen of Ser79 mediate hydrogen bonds with the amide nitrogen and carbonyl oxygen of Phe31, respectively. To verify the conformation of the secondary cleavage site loop, we solved the structure of isolated **M2** at pH 8.3 and refined it to 1.3 Å resolution (Table S1). Indeed, this region showed the same conformation in isolated and furin-bound **M2** (Figs. 2B and S7A, B).

In conclusion, the binding of the PC1/3-prodomain to furin is mediated by exosite (globular part) as well as substrate-like interactions (C-terminus). The structural data suggest a specific destabilization of the secondary cleavage site loop at acidic pH.

**M2** includes three Arg-Ala mutations to prepare a stable prodomain:furin complex. To confirm structural similarity between **M2** and the wild-type PC1/3-prodomain, we performed NMR-experiments in solution at pH 7.4. For an NMR analysis of the PC1/3-prodomain in solution, uniform $^{15}$N and $^{13}$C/$^{15}$N labeling of the recombinant proteins was applied, and triple-resonance 2D and 3D spectra were used to obtain a complete sequence-specific resonance assignment as a basis for structural studies (Fig. S8). The protein is under this condition folded, and the predicted secondary structure based on the chemical shift assignment agrees with the secondary structure in the crystal structure (Fig. S9A). Whereas the crystal structure reported a β-strand around residues 75-80, the NMR data of that region did not reach the threshold to predict a β-strand (Fig. S9A). It could only hint to some population of a β-strand in that region. We measured $^{15}$N{$^{1}$H} heteronuclear NOE data, a sensitive indicator for dynamics in the subnanosecond timescale (Fig. S9B). Surprisingly, residues 75–80 seem rather rigid (at least in the sub-nanosecond timescale), but Asn74 and His75 show some dynamics.

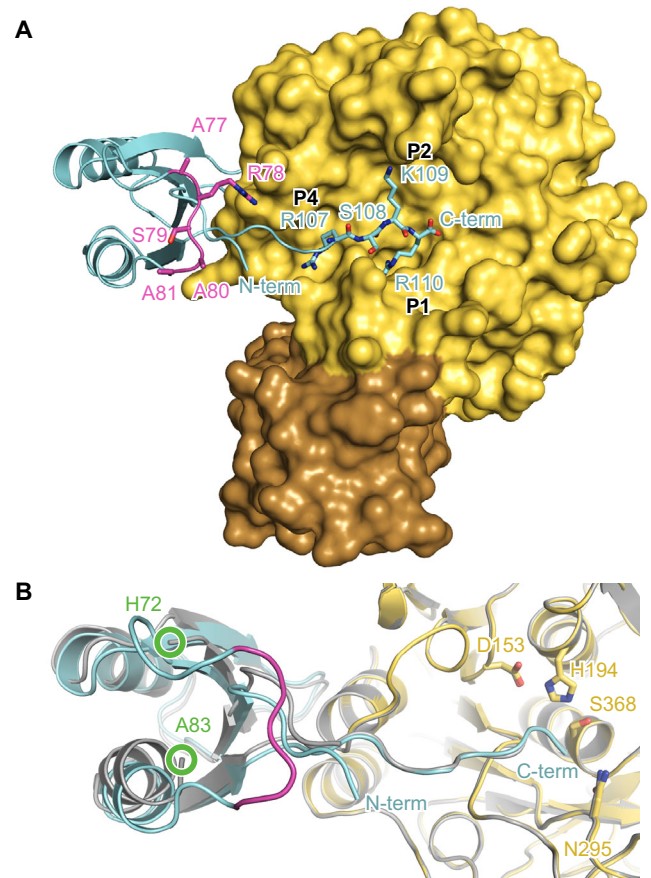

**A**

**B**

**Fig. 1 | Crystal structures of the PC1/3-prodomain bound to furin.** Furin is shown in standard orientation. **A** The molecular surface of furin bound to **M2** (M = mutant) is colored in yellow (catalytic domain) and brown (P-domain). The prodomain is shown as cartoon representation (cyan). The C-terminal residues 107-110 (P1-P4) are shown as stick model. The secondary cleavage site loop is marked in magenta. Note that this loop of **M2** carries the sequence 77-ARSAA instead of the wild-type sequence 77-RRSRR. **B** Superposition of **M2**:furin (pH 7.0) with **M5**:furin (pH 5.9). The cartoon representations of **M2**:furin and **M5**:furin are colored according to A) and in gray, respectively. The active site residues are given as stick model. The region between His72 and Ala83 was flexible and thus was not built in the model. The positions of this chain break are marked by green circles.

Unambiguous NOEs within the critical loop region were extracted from the acquired 2D and 3D NOESY spectra (Figs. 2C and S10, and Table S2). The measured NOEs of residues His75, Arg77 (Ala77 in **M2**), Ser79, Arg80 (Ala80 in **M2**) and Ser72 agree with the X-ray structure. For instance, we found NOE contacts between the amide protons of Ser79 and Phe31, which is in excellent agreement with the interaction of these residues found in the X-ray structure.

### His72 forms a pH-sensitive hydrogen bond to backbone amide atoms

In the isolated structure of **M2** and in the complex structure with furin, we found an unusual interaction of the His72 side chain with the amide nitrogen atoms of Asn74 and His75 (Figs. 2B and S7B). The donor-hydrogen-acceptor angle towards Asn74 (169°) indicates a stronger contribution than the less optimal angle towards His72 (155°) as determined from the isolated **M2** structure. The distances between the amide nitrogen atoms and $N_\delta$ of His72 are in both cases 3.1 Å. This observation is supported by the NMR-data, showing NOEs between His72-Hε1 and Asn74-$H^N$, Hβ2, Hβ3, as well as His72-Hε1 and His75 $H^N$ (Fig. 2C and Table S2). A 2D $^{15}$N-HMBC experiment revealed the most common neutral Nε2H tautomeric state for His72, His75, and H75

(Fig. S11). His72 can only act as a hydrogen bond acceptor in a deprotonated state. Thus, protonation of His72 at pH 6.0 will change a productive interaction to an unfavored contact, probably destabilizing the conformation of the secondary cleavage site loop. Consistent with this observation, the last amino acid with visible electron density at the N-terminal end of the secondary cleavage site loop in **M5**:furin was His72 (Fig. S4B).

Based on these findings, we conclude that the protonation of His72 would prevent its sidechain-mainchain interactions and destabilize the conformation of the secondary cleavage site loop.

To investigate the pK$_a$-values of the histidine residues close to the secondary cleavage site loop of the isolated PC1/3-prodomain, we measured 2D $^{15}$N-HMBC spectra of $^{15}$N-labeled wild-type protein in dependence of the pH. Because of the inherent instability and partial unfolding of the protein at room temperature below pH 7.4 (see above), pH-dependent experiments were performed at ~5 °C (Fig. S12A). Even under optimized conditions, we observed unfolding of the protein at pH 6.0; thus, the pH range from 6.3 to 7.4 was evaluated further. We followed the chemical shifts of the non-protonated nitrogens (Nδ1 in case of the Hε2H tautomer), because they show the most dramatic chemical shift deviations upon protonation, and they report protonation directly and in a predictable manner (>240 ppm if unprotonated and ~180 ppm if positively charged). A fit of the data under the assumption that at pH 2.0 all histidine residues are protonated and that the $^{15}$N chemical shift will be then 176 ppm revealed pK$_a$-values of 4.82, 5.03, and 5.60 for His72, His75, and His85, respectively (Fig. S12B)[33,34].

### Engineering of a stable inhibitor:enzyme complex by exchange of His72

According to the observed interaction scheme, we hypothesized that an introduction of an asparagine should be able to substitute His72. However, unlike histidine, asparagine is not protonated and can also act as hydrogen acceptor at pH 6.0. Thus, we expected a stabilization of the secondary cleavage site loop for a His72Asn mutant at pH 6.0. Aspartate also acts as hydrogen acceptor at pH 6.0 and pH 7.4 and should be a suitable substitute for His72 as well. To test this hypothesis, we performed limited proteolysis with the His72Asn (**M6**) and His72Asp (**M7**) mutants of the PC1/3-prodomain. Indeed, we observed reduced cleavage of **M6** and **M7** by furin at pH 6.0 in limited proteolysis experiments compared to the wild-type (Fig. S13A–C). The secondary cleavage of **M6** and **M7** by furin at pH 7.4 was not affected. This finding supports a crucial role of the histidine-backbone amide interactions as pH-responsive element in the PC1/3-prodomain. The His72Asp mutation also introduces a negative charge, which apparently did not influence cleavage of the secondary cleavage site loop compared to the uncharged His72Asn mutation. In the literature, a His72Leu mutation has been reported to increase the stability of the PC1/3-prodomain and abolish auto-activation of PC1/3 at pH 6.0[27]. Based on our structures, we hypothesized, however, that a leucine cannot maintain the same side chain interactions as a histidine at this position and probably causes additional structural effects. Due to its different stereochemical properties, a leucine residue should not fit easily at this position. To test this hypothesis in biochemical assays, we generated a His72Leu mutant of the PC1/3-prodomain (**M8**). For **M8** we indeed observed a largely reduced cleavage by furin at pH 7.4 compared to the wild-type protein (Fig. S13D). At pH 6.0, **M8** was cleaved with comparable efficiency as found for pH 7.4. This result shows that the His72Leu mutation interferes with secondary cleavage of the prodomain in two different ways: (a) by reducing the cleavage rate in general (b) by impairing its pH-sensitivity. In fact, processing of the secondary cleavage site of **M8** by furin was even lower at pH 6.0 compared to the wild-type prodomain at pH 7.4 (Fig. S13A, D).

We also investigated the influence of these mutations on the structural stability of the mutants **M6**, **M7**, and **M8** in nano-DSF

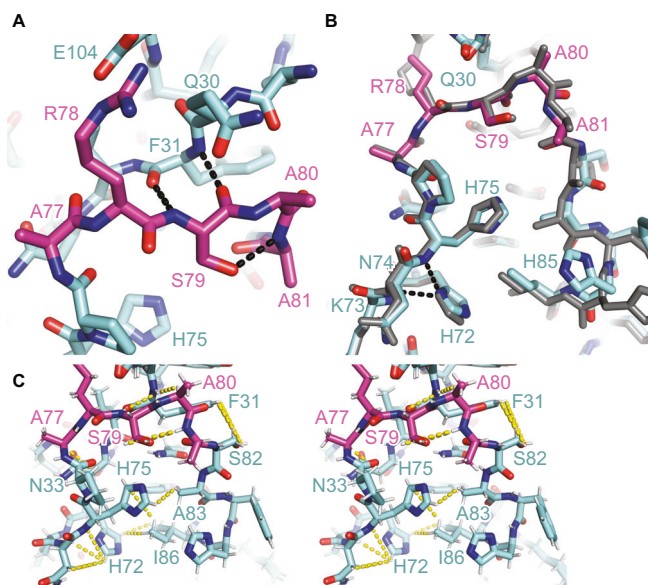

**Fig. 2 | Secondary cleavage site loop of the PC1/3-prodomain. A** Interactions of the secondary cleavage site loop (magenta) of **M2** (M = mutant) bound to furin. The protein is shown in stick representation, and hydrogen bonds are highlighted by black dashes. **B** Structural alignment of the secondary cleavage site loop (magenta) and flanking regions of isolated **M2** (colored) and of **M2** (gray) bound to furin. For the structure of isolated **M2** only the positions of $C_\alpha$, $C_\beta$, and $C_\gamma$ of the side chain of Arg78 were build. Gln30 was modeled in two conformations. Hydrogen bonds between the side chain of His72 and the backbone nitrogens of Asn74 and His75 are highlighted with black dashes. **C** Stereo view of the secondary cleavage site loop (magenta) and flanking regions of isolated **M2**. Nuclear Overhauser effects (NOEs) measured by NMR spectroscopy (Table S2) are highlighted by yellow dashes. The NOE data suggest flipping of the His75 imidazole ring in solution.

experiments. Surprisingly, we observed a drop of the $T_m$ of the mutants at pH 6.0 compared to pH 7.4, similar to the wild-type protein (Table 2). Hereby, **M8** also showed an increased $T_m$ at pH 6.0 and pH 7.4 compared to the other mutants. An important difference between these nano-DSF measurements and the limited proteolysis assays is, however, the presence of furin. Thus, we repeated the nano-DSF experiments with the preformed furin:prodomain complexes (Table 2). To prevent degradation of the prodomain by furin, we introduced the His72Leu mutation in the secondary cleavage site mutants **M2** and **M5**, resulting in the combination mutants **M9** and **M10**. Because nano-DSF measurements rely on tryptophan fluorescence, the signal from furin (11 Trp-residues) is largely dominant over the PC1/3-prodomain (2 Trp-residues). For all complexes, we observed melting curves with one transition point only (Fig. S3C, D). It is noteworthy that the $T_m$ of isolated furin was also reduced from $65.6 \pm 0.1\,°C$ at pH 7.4 to $63.1 \pm 0.0\,°C$ at pH 6.0. The binding of the prodomains always increased the structural stability of furin, as indicated by increased $T_m$-values at pH-values 6.0 and 7.4. In general, the pH-dependent differences of the $T_m$-values measured for prodomain:furin complexes were smaller compared to the isolated prodomains. The His72Leu combination mutants revealed the highest temperature stabilities with $T_m$-values of $72.5 \pm 0.0\,°C$ at pH 7.4 and $71.8 \pm 0.0\,°C$ at pH 6.0 for **M9** and $75.5 \pm 0.1\,°C$ at pH 7.4 and $73.4 \pm 0.1\,°C$ at pH 6.0 for **M10**.

### Combination of secondary cleavage site and H72 mutations inverted the pH-dependent affinity profile of the PC1/3-prodomain

Our findings strongly suggest a dependency between the structure of the secondary cleavage site loop of the PC1/3-prodomain and its inhibitory potency. Thus, we also aimed at the determination of $K_i$-

values at pH 6.0. At pH 6.0 and 37 °C, however, we could not fit the kinetic data with a competitive inhibition model. This could be explained by a partial unfolding of the isolated prodomain under these conditions, as indicated by the relatively low $T_m$-values. To stabilize the PC1/3-prodomain we used optimized assay conditions with 300 mM NaCl and measured at 25 °C. For **M5** we observed a ~2.5-fold decreased affinity at pH 6.0 ($K_i = 0.35 \pm 0.02\,nM$) compared to pH 7.4 ($K_i = 0.139 \pm 0.008\,nM$, (Table 2 and Fig. S14). The pH-dependent drop of affinity of **M10** (equivalent to **M5** with His72Leu mutation) was largely reduced, as indicated by $K_i$-values of $0.14 \pm 0.01\,nM$ and $0.101 \pm 0.007\,nM$ at pH 6.0 and pH 7.4, respectively. Interestingly, for **M2** we observed similar $K_i$-values at pH 6.0 and pH 7.4 ($0.88 \pm 0.03\,nM$ and $0.73 \pm 0.02\,nM$, respectively). Surprisingly, the affinity of the combination mutant **M9** (equivalent to **M2** with His72Leu mutation) was even higher at pH 6.0 ($K_i = 0.35 \pm 0.01\,nM$) compared to pH 7.4 ($K_i = 0.62 \pm 0.03\,nM$).

### Engineering of a highly furin-specific prodomain-based inhibitor

Nanobodies inhibit furin with excellent specificity but with lower affinity compared to the PC1/3-prodomain. Comparing the structures of furin in complex with **M2** and of furin in complex with the nanobody Nb14[18], simultaneous binding of both proteins should be possible (Fig. 3). Thus, we hypothesized that a fusion protein of the nanobody Nb14 and of the PC1/3-prodomain should result in an improved inhibitor with very high affinity and specificity. According to the structures, the N-terminus of the prodomain is only 27 Å away from the C-terminus of the nanobody. In principle, 8 amino acids are sufficient to bridge this distance and link the two proteins. We tested different linker lengths by energy minimization in COOT. Finally, 11 amino acids were the minimal length to achieve good geometric properties. The linker sequence was chosen to avoid charged amino acids that are prone to degradation by various proteases. On the other hand, amino acid residues were chosen to fit the surface characteristics of the nanobody and of the PC1/3-prodomain at the respective positions. This feature might allow interactions of the linker with the proteins, which might reduce its flexibility and thus its accessibility for proteolytic attack. Using Alphafold3 we predicted the structure of the complex between the nanobody fusion protein of **M9** and furin (referred to as **F1**, Figs. 3 and S15). In this structure, the linker shows the optimal length to connect the fusion partners and maintaining their binding mode. In addition, we also created a fusion protein of **M5** and furin (referred to as **F2**).

A competitive inhibition mechanism of **F2** was indicated by a linear relationship of $IC_{50}$-values in dependence on substrate concentration/$K_m$-value under tight binding conditions (Fig. S16A). The $K_i$-values of **F1** and **F2** (150 mM NaCl, pH 7.4, 37 °C) were $1.6 \pm 0.4\,pM$ and $1.3 \pm 0.3\,pM$, respectively (Table 3 and Fig. S16B, C). To test whether the binding mode indeed involves cooperative binding of the nanobody and prodomain parts of the fusion protein, we examined binding to furin$^{Thr562Arg}$. Thr562 is a non-conserved residue within the PC protease family, and its mutation to Arg reduced the affinity of furin to the nanobody but fully maintained the catalytic activity[18]. Indeed, the affinity of **F1** and **F2** for furin$^{Thr562Arg}$ compared to wild-type furin dropped by more than two orders of magnitude as indicated by the $K_i$-values of $880 \pm 60\,pM$ and $215 \pm 16\,pM$, respectively (Table 3 and Fig. S16D). This finding and the competitive inhibition mechanism support simultaneous binding of both fusion partners.

Next, we tested the specificity of the fusion proteins for PC7 and PC5/6, whereby PC7 shows the lowest and PC5/6 shows high sequence homology to furin among the PC-family. We determined $K_i$-values of **F1** ($440 \pm 43\,pM$) and **F2** ($1200 \pm 110\,pM$) for PC7 showing a stronger inhibition of furin by ~275- and ~900-fold (Table 3 and Fig. S16E). PC5/6 was inhibited by **F1** and **F2** with $K_i$-values of $32,600 \pm 2400\,pM$ and $36,800 \pm 2700\,pM$ (Table 3 and Fig. S16F), respectively. This indicates ~23,000- and ~25,000-fold stronger inhibition of furin by **F1** and **F2**

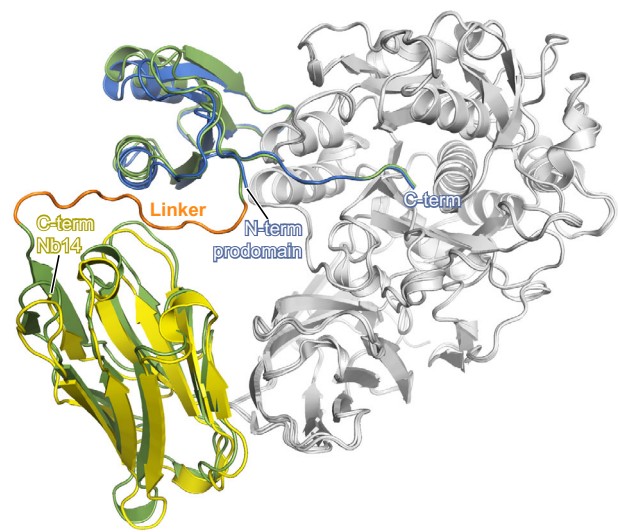

**Fig. 3 | Structure-based engineering of fusion proteins of the PC1/3-prodomain with the furin-inhibiting nanobody Nb14.** Superposition of the structures of furin (gray) in complex with **M2** (M = mutant, blue, experimental structure, this work), Nb14 (yellow, experimental structure, 5JMO[18]) or **F1** (F = fusion protein, green, predicted by Alphafold 3[60]). The N- and C-termini of the PC1/3-prodomain, as well as the C-terminus of the nanobody, are highlighted. The linker that connects these termini in the fusion protein is colored in orange.

**Table 3 | $K_i$-values of nanobody-PC1/3-prodomain fusion proteins F1 and F2**

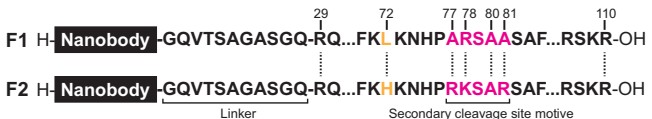

| | No. | Furin | Furin$^{T562R}$ | PC7 | PC5/6 |
|---|---|---|---|---|---|
| $K_i^a$(pM) | F1 | 1.6 ± 0.4 | 880 ± 60 | 440 ± 43 | 36,800 ± 2700 |
| | F2 | 1.3 ± 0.3 | 215 ± 16 | 1200 ± 110 | 32,600 ± 2400 |

$^a$pH 7.4, 150 mM NaCl, 37 °C.

compared to PC5/6. In conclusion, our prodomain-nanobody fusion proteins are highly specific furin inhibitors.

### Anti-viral activity of prodomain-nanobody fusion proteins

To test the biological activity of our fusion proteins, we tested their antiviral activity in cell culture-based assays. For this purpose, we infected A549 human lung cancer cells with the influenza A virus strain H7N7/SC35M and determined the viral titer in the supernatant at 16, 24, 48, and 72 h post-infection (Fig. 4). One hour after infection we added the prodomain mutants, the fusion proteins or the nanobody as control protein to the medium. As additional control we also tested a fusion protein of the nanobody and the wild-type PC1/3-prodomain. Untreated cells and cells treated with the previously described furin inhibitor MI-1148[35] served as additional controls. We observed a dose-dependent inhibition of viral replication for fusion protein **F1** with a 10,000-fold reduction in final virus titer at 72 h post-infection at 10 μM and a 100-fold reduction at 5 μM concentration. A slight reduction in virus titer was observed for **F2** at 48 h post-infection at 10 μM concentration. In comparison, no antiviral activity was observed for the isolated wild-type PC1/3-prodomain, the isolated mutant prodomains **M5** and **M9** (prodomain parts of **F1** and **F2**), the isolated nanobody, as well as a fusion protein of the nanobody and the wild-type PC1/3-

prodomain. MI-1148 completely blocked replication of H7N7/SC35M at a concentration of 1 μM. We also determined the cleavage of the viral envelope protein hemagglutinin (HA) in the cells by western blot in presence of the prodomain fusion proteins **F1** or **F2** or MI-1148 as a control (Fig. 5A). Efficient cleavage of the precursor HA0 into HA1 and HA2 was observed in untreated infected cells. In agreement with the reduction of viral titers by **F1**, cleavage of HA0 into HA1 and HA2 was reduced in a dose-dependent manner and fully prevented in the presence of 10 μM **F1**, indicating a strong inhibition of PC-dependent HA cleavage. HA0 cleavage was also slightly reduced in **F2** (10 μM) treated cells compared to untreated cells. In the presence of MI-1148, HA cleavage was blocked, and only the precursor HA0 was detected.

The different antiviral activities (activity in the cells) of **F1** and **F2** are remarkable because the in vitro inhibitory potency of **F2** was stronger compared to **F1**. This finding suggests that the prodomain part is important for the biological activity of the fusion proteins, i.e., the mutations of the secondary cleavage site and of the pH-sensing His72. To test this hypothesis, we co-incubated the inhibitory proteins in the medium together with A549 human lung cancer cells. Then, we monitored the inhibitory activity of the isolated prodomains and of the prodomain fusion proteins after 24 h and 72 h incubation (Fig. 5B). For the wild-type prodomain, no inhibitory activity was measured after 24 h, consistent with processing and inactivation of the prodomain by cellular PCs. In contrast, for **M5** we observed full inhibition at 24 h and week inhibition until 72 h. For the combined mutant **M10** (M5 with His72Leu mutation), we observed even after 72 h ~70% inhibition. Interestingly, the remaining inhibition of **M9** after 72 h was almost 100%. For the fusion proteins **F1** and **F2** we measured strong inhibition over 72 h. Nonetheless, we could determine an approx. two-fold lower decline of the inhibition for **F1** in correlation with the findings for **M5** and **M10**. The stable inhibition observed for the fusion proteins indicates that the linker between the prodomain and the nanobody is apparently relatively resistant against proteolytic degradation. This observation is supported by the absence of degradation products in the range of ~15 kDa in the cell culture supernatants (Fig. S17).

## Discussion

During the canonical auto-activation of the PCs, the prodomains are autocatalytically cleaved and inactivated in a pH-dependent manner[1]. Proteolysis occurs in the secondary cleavage site loop of the prodomains and is initiated by trafficking of the proteases. After synthesis of the pro-PCs in the endoplasmic reticulum at a pH of 7.4, the lower pH of ~6.0 at the late Golgi triggers the auto-activation process[36]. So far, the inherent instability of PC-prodomains has prevented their use as inhibitory proteins. Influenza virus HA is typically cleaved in secretory compartments at pH values of 5.5–6.0. Under these conditions, the wild-type PC1/3-prodomain could be readily inactivated by endogenous furin, explaining the lack of any antiviral activity.

According to our structural and biochemical data, two main determinants of the pH-sensitivity of the PC1/3-prodomain are (1) the multi-basic motif of the secondary cleavage site loop and (2) His72. In previous studies, Arg→Ala point mutations at the secondary cleavage site motif either did either not affect or slightly increased its affinity to furin[31]. In fact, we measured an almost identical $K_i$ value of **M1** (Arg78Ala) as described in the literature (0.59 nM[31]). We identified a double (**M5**) and a triple-mutant (**M2**) of the secondary cleavage site motif, maximizing the affinity and the stability of the PC1/3-prodomain.

In the crystal structure of the isolated PC1/3-prodomain and in complex with furin, the side chain imidazole ring of His72 mediates a hydrogen bond to a backbone amide of the secondary cleavage site loop. Protonation of the imidazole ring at acidic pH prevents the hydrogen bond acceptor function of His72 and thus destabilizes the secondary cleavage site loop conformation. This loop is probably more susceptible to proteolytic attack in an unstructured state, explaining

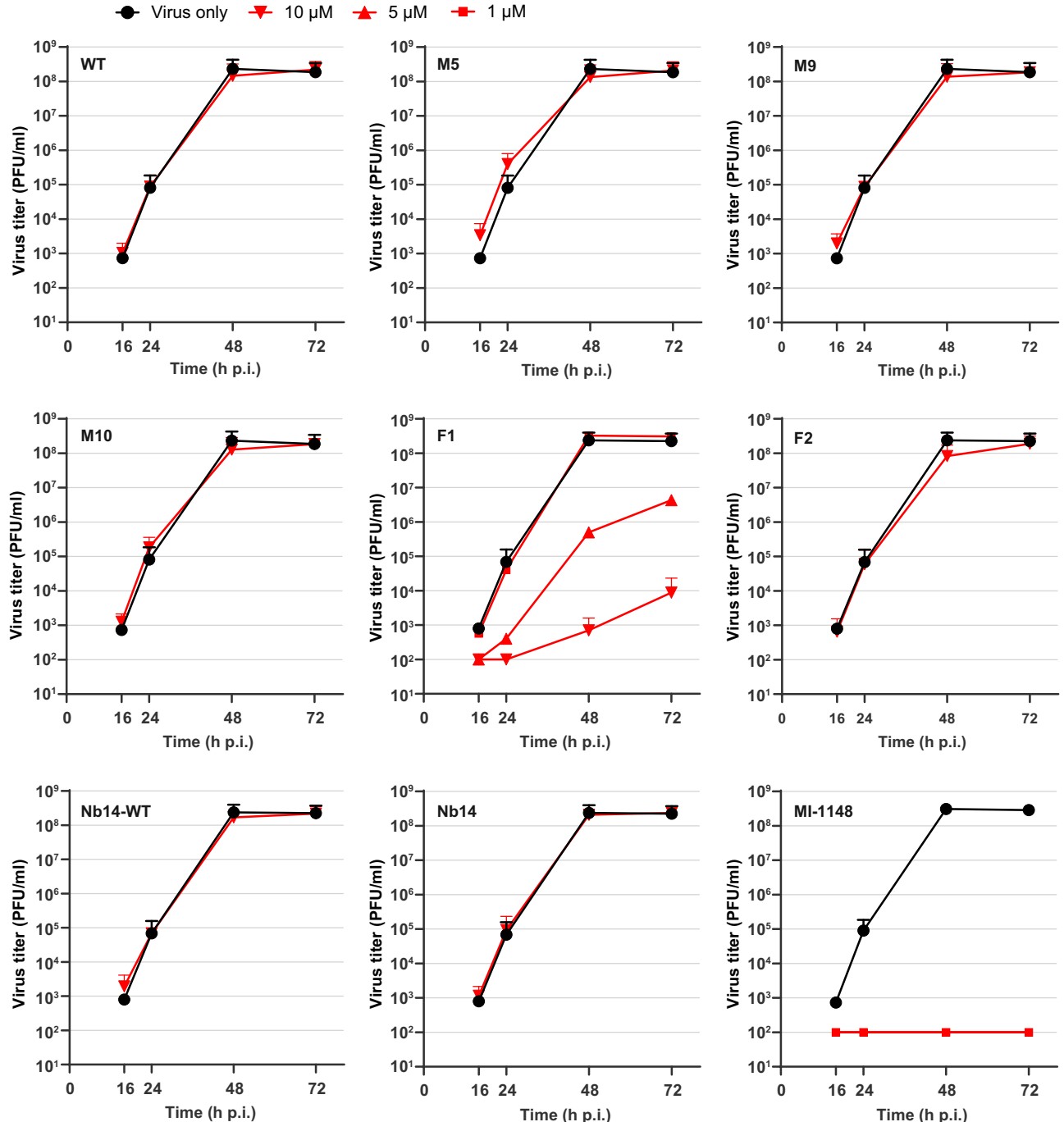

**Fig. 4 | Inhibition of multicycle replication of influenza A virus H7N7/SC35M in inhibitor-treated A549 human lung cells.** Cells were inoculated with virus at an MOI of 0.0001 for 1 h, washed, and incubated in the presence of 10, 5, or 1 μM of the indicated inhibitors for 72 h. As a control, cells were infected with H7N7/SC35M in absence of an inhibitor. At 16, 24, 48, and 72 h post-infection (p.i.), cell supernatants were collected and viral titers were analyzed by plaque assay. Data are mean values + SD of three independent experiments ($n = 3$).

the increased furin cleavage of the PC-prodomain at acidic pH. A mutation of His72 to leucine was reported to disrupt the pH-sensing function of the PC1/3-prodomain[27]. We could show that this mutation resulted in a global stabilization of the PC1/3-prodomain as indicated by higher melting temperatures at pH 6.0 and pH 7.4. Due to its different stereochemical properties, the leucine side chain might fit better to the hydrophobic core of the prodomain. A combination of the secondary cleavage site triple mutation (77)A-R-S-A-A (**M2**) with the His72Leu mutation resulted in a largely stabilized inhibitory PC1/3-prodomain (**M9**). In fact, the melting temperatures of the furin:**M9**

complex at pH 6.0 and pH 7.4 were largely similar. The affinity of **M9** to furin was even higher at pH 6.0 than at pH 7.4.

Interestingly, we observed a relatively low $pK_a$-value of 4.82 for His72 of the isolated PC1/3-prodomain. This might be different for the furin-bound prodomain due to the influence of the interaction with the catalytic domain on its biochemical properties.

To achieve both high affinity and selective furin inhibition, we fused our optimized PC1/3-prodomain variants to the C-terminus of a furin-specific nanobody. These fusion proteins indeed revealed very high affinity to furin and very high specificity, especially in comparison

**A**

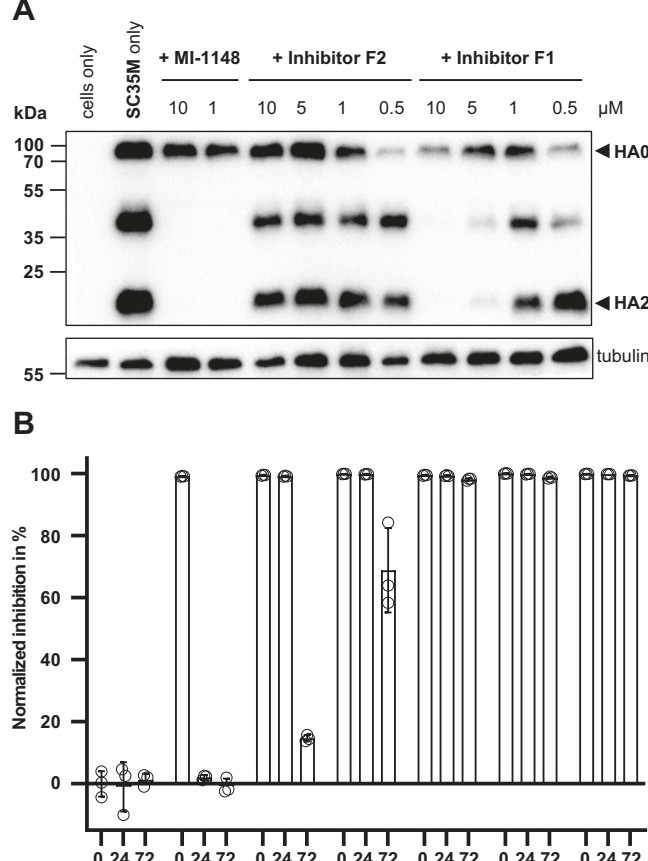

**B**

**Fig. 5 | Inhibition of proteolytic activation of influenza A virus H7N7/SC35M and stability of the PC1/3-prodomain in cell culture. A** Analysis of HA cleavage. A549 cells were infected with H7N7/SC35M at a MOI of 0.1. After a 1 h incubation, the inoculum was removed and cells were further incubated in the presence of inhibitors for 24 h. Cell lysates were subjected to SDS-PAGE and Western blot analysis. The precursor HA0 and mature cleavage products HA1 and HA2 were immunochemically detected with an H7 hemagglutinin-specific antibody. Tubulin was used as a loading control. The Western blot shown is representative of three independent experiments (n = 3). (M = mutant, F = fusion protein) (**B**) Stability of the wild-type PC1/3-prodomain (WT), PC1/3-prodomain mutants (M), and fusion proteins (F) in A549 lung cancer cell cultures. The inhibition is given in % as the ratio between the activity measured with conditioned medium supplemented with the test proteins and cell culture medium. Conditioned medium supplemented with buffer only served as a negative control (Control). The experiment was performed in triplicates (n = 3), Mean values and standard deviations are shown; open circles represent individual replicates.

to PC5/6. This is a remarkable finding because PC5/6 shows an especially high sequence conservation to furin among the PC-family members[13]. In principle, this strategy might be applied to develop highly specific inhibitors against any PC family member. Our optimized PC1/3-prodomain variant **M9** might serve as a general anchoring platform for such fusion proteins. The preformed complex between the prodomain and the PC of interest could be directly used to identify suited nanobodies with display technologies[37].

**F1** and **F2** showed antiviral activity against the avian influenza virus strain H7N7/SC35M. Hereby, the activity of **F1** was much stronger compared to **F2**, although both fusion proteins showed comparable affinities and specificities. Interestingly, we observed a higher structural stability of **M9** (the prodomain of **F1**) compared to **M5** (the prodomain part of **F2**) in biochemical assays. Thus, the stability of the prodomain part might be crucial for an antiviral effect of the fusion

proteins. This property could be especially important for the half-life of the fusion proteins after cellular uptake. In the secretory compartments, various proteases with broad specificity are present (e.g., the cathepsins)[38,39], which might digest more flexible parts in the fusion proteins (e.g., the linker region or flexible loops). In this context, the pH stability and the increased global structural stability of **M9** (H72L mutation in combination with (77)A-R-S-A-A secondary cleavage site sequence) are probably important. Without these mutations, the pH of ~5.5 as found in endosomes largely destabilizes the PC1/3-prodomain. The prodomain part and especially the secondary cleavage site loop of **F2** (**M5**) might be partially unfolded and thus inactivated through proteolysis by cathepsins and legumain[40]. A more efficient cellular uptake of **F1** compared to **F2** might also contribute to a higher biological activity.

The higher antiviral activity of the peptidic inhibitor MI-1148 compared to **F1** might be explained by different specificities of these inhibitors for different PCs. **F1** inhibits furin ~25,000-fold stronger compared to PC5/6, while MI-1148 showed similar inhibition for both PCs[14]. HA of highly pathogenic avian influenza virus (HPAIV) strains, however, is activated by furin and PC5/6[41]. Specific inhibition of furin only cannot result in a complete inhibition of SC35M replication, which is in agreement to the results obtained for **F1**.

Due to their very high affinities, **F1** and **F2** might be applied for sensitive detection of active furin in cells or biological samples. These protein-based inhibitors can be easily fused to an enzyme (e.g., horse radish peroxidase or luciferase), a fluorescent protein, or an antibody F$_c$-domain to facilitate compatibility with common detection approaches.

## Methods
### Protein expression and purification
Shortly, Furin[5,32,42], Furin[T562R 18], PC5/6[42], and PC7[42] were expressed in HEK293S cells (ATCC CRL3022, not authenticated) as soluble proteins and purified from conditioned medium. All proteins were initially subjected to immobilized metal affinity purification (IMAC). Furin and Furin[T562R] were further purified by inhibitor-based affinity purification[43]. Finally, the buffer of Furin and Furin[T562R] was exchanged by ultrafiltration (Amicon Ultra 15 Centrifugal Filter Ultracel 30 K, Merck Millipore) to storage buffer (10 mM Hepes/NaOH, pH 7.4, 100 mM NaCl, and 2 mM CaCl$_2$). PC7 was activated in vitro using thermolysin[42]. PC5/6 and PC7 were subjected to a final gel permeation chromatography step (Superdex 200 10/300 GL column, GE Healthcare) coupled to a chromatography system (Aekta Purifier with Unicorn 5.31 software, GE Healthcare) with storage buffer. Details of the cloning procedures, expression, and purification are provided as supplementary information.

Synthesized nucleic acid sequences of the PC1/3 prodomain constructs (wild-type, **M1-M5, M9,** and **M10**) and of the nanobody-PC1/3-prodomain fusion constructs were cloned into the pET28a vector (Novagen) by Geneart (Thermo Fisher Scientific). PC1/3 prodomain constructs **M6, M7,** and **M8** (His72 mutants) were generated by site-directed mutagenesis. For protein purification N-terminal His$_6$- (used for wild-type, **M2, M3, M9,** and **M10**) or His$_6$-mNeongreen- (used for wild-type, **M1, M4-M8**) tags were fused to the prodomains via the linker sequence SGTENLYFQG that includes the tobacco etch virus (TEV)-protease cleavage site (ENLYFQ↓G). The expression strategy with the N-terminal His$_6$-tag revealed higher protein yields and was applied when higher amounts of protein were needed. After cleavage of the tag, all proteins contained a single N-terminal glycine residue as cloning artifact, followed by the amino acids 28–110 of human PC1/3 (PCSK1, UNIPROT-ID P29120). We also produced a His$_6$-tagged variant of **M5** containing the sequence GHHHHHHSGHM N-terminal to the amino acids 28–110 of human PC1/3, which was only used for crystallization (see below). Details of the cloning procedures are provided as supplementary information.

The proteins were expressed in *E. coli* BL21 (DE3) grown in lysogeny broth medium (Carl Roth, Karlsruhe, Germany) supplemented with 7.5 µg ml$^{-1}$ kanamycin at 37 °C. At $OD_{600} = 0.4$ the cells were cooled to 26 °C and protein expression was induced by addition of 1 mM isopropyl β-d-1-thiogalactopyranoside at $OD_{600} = 0.8–1.0$. After 4 h, cells were harvested and stored at −20 °C until purification. Cell pellets were resuspended in lysis buffer (100 mM Tris/HCl, pH 8.0, 500 mM NaCl), supplemented with 1 mg ml$^{-1}$ lysozyme, incubated for 1 h at 4 °C, and lysed by sonication. For immobilized metal affinity chromatography (IMAC), the supernatant was supplemented with 10 mM imidazole and incubated with Ni-NTA Superflow resin (Qiagen, Hilden, Germany) for 30 min at 4 °C. The resin was transferred to a gravity flow column and washed 2x with 10 bed volumes of lysis buffer containing 10 mM imidazole and with 10 bed volumes of lysis buffer containing 20 mM imidazole. The proteins were eluted with lysis buffer containing 250 mM imidazole. Prior to cleavage of the tags, the buffer was exchanged to 10 mM Hepes/NaOH, pH 7.4, 300 mM NaCl by dialysis (Slide-A-Lyzer, 3.5 kDa, Thermo Fisher Scientific). The tags were cleaved by addition of TEV-protease in a ratio of 1:100 (w/w, His$_6$-tag) or 1:20 (w/w, His$_6$-mNeongreen-tag) and incubation at room temperature overnight. The cleavage reaction was loaded onto Ni-NTA resin. The tag-less target protein was collected in the flow-through as well as after elution with lysis buffer containing 10 mM imidazole. The PC1/3-prodomain variants were concentrated by ultrafiltration (Amicon Ultra 15 Centrifugal Filter Ultracel 3 K, Merck Millipore). Final purification was performed by gel filtration chromatography (HiLoad 16/60 Superdex 75 pg column, Amersham Biosciences) coupled to a chromatography system (Aekta Purifier with Unicorn 5.31 software, GE Healthcare) in 10 mM Hepes/NaOH, pH 7.4, and 300 mM NaCl.

For NMR-spectroscopy the N-terminal His$_6$- ($^{15}$N-label) or His$_6$-mNeongreen-taged ($^{15}$N/$^{13}$C-label) wild-type PC1/3-prodomain was expressed in minimal medium containing 0.5% (w/v) glucose, 50 mM NH$_4$Cl, 2 mM MgCl$_2$, 50 µM FeCl$_3$, 20 µM CaCl$_2$, 10 µM MnCl$_2$, 10 µM ZnCl$_2$, 2 µM CoCl$_2$, 2 µM CuCl$_2$, 2 µM NiCl$_2$, 2 µM Na$_2$MoO$_4$, 2 µM Na$_2$SeO$_3$, 2 µM H$_3$BO$_3$, 25 mM Na$_2$HPO$_4$, 25 mM KH$_2$PO$_4$, 5 mM Na$_2$SO$_4$ as described above. For $^{15}$N- and $^{13}$C-labeling, $^{15}$NH$_4$Cl (Cambridge Isotope Laboratories) and $^{13}$C$_6$ glucose (Cambridge Isotope Laboratories) were used, respectively. N-terminal His$_6$-tagged PC1/3-prodomain formed non-classical inclusion bodies (IBs) in minimal medium. Cell pellets were resuspended in lysis buffer (100 mM Tris/HCl, pH 8.0, 500 mM NaCl), supplemented with 1 mg ml$^{-1}$ lysozyme, incubated for 1 h at 4 °C, and lysed by sonication. Cell lysates were centrifuged at $17500 \times g$, and the pellet fraction (containing the IBs) was washed in lysis buffer. The IBs were dissolved overnight at 4 °C in a mixture of 62.5% lysis buffer and 37.5% 50 mM Tris/HCl, pH 9.0, 8 M urea. For IMAC the solubilization mixtures were supplemented with 10 mM imidazole. The isotope labeled protein variants were purified according to the procedure described above, and the buffer was exchanged to 50 mM Na$_2$HPO$_4$/NaH$_2$PO$_4$, 150 mM NaCl, pH 7.4 by dialysis. The protein concentration was adjusted to 200 µM, and 7.0% (v/v) D$_2$O was added prior to NMR measurements (see below).

The nanobody fusion proteins were expressed with an N-terminal pelB secretion-signal sequence and an His$_6$-tag fused to the N-terminus of the nanobody via the linker sequence SGTENLYFQG. The PC1/3-prodomain was fused C-terminal to the nanobody using the linker sequence GQVTSAGASGQ (Table 3). The fusion proteins were expressed in terrific broth (Carl Roth, Karlsruhe, Germany) supplemented with 7.5 µg ml$^{-1}$ kanamycin at 37 °C. At $OD_{600} = \sim 1.0$ the cells were cooled to 26 °C, and protein expression was induced by addition of 1 mM isopropyl β-d-1-thiogalactopyranoside at $OD_{600} = 10$. After overnight incubation, cells were harvested and stored at −20 °C until purification. Preparation and solubilization of the non-classical IBs, protein purification, and tag removal were performed as described above.

The nanobody Nb14 was expressed as a reverse fusion construct with the PC1/3-prodomain **M2**. For this purpose, the construct contained the N-terminal pelB secretion signal, a His$_6$-tag, the PC1/3-prodomain, and the nanobody at the C-terminus. Between Glu106 PC1/3-prodomain and Gln1 of the nanobody, the linker sequence NLYFQGS was placed, containing the TEV-cleavage site. Expression, purification, and tag removal were performed as described above.

## Limited proteolysis assays

Proteolysis of PC1/3 prodomain variants at pH 6.0 (50 mM Mes/NaOH, pH 6.0, 150 mM NaCl, 2 mM CaCl2) and pH 7.5 (50 mM Hepes/NaOH, pH 7.5, 150 mM NaCl, 2 mM CaCl2) was investigated by incubation with different concentrations of recombinant furin at 37 °C. **M1-M5** and the wild-type protein (control) were incubated at a concentration of 0.189 mg ml$^{-1}$ with furin in a 1.5:1 molar ratio. **M6-M8** and the wild-type protein (control) were incubated at a concentration of 0.189 mg ml$^{-1}$ with furin in a 15:1 molar ratio. PC1/3-prodomain without furin was used as control. Progress of proteolysis was investigated after distinct time points via SDS-PAGE.

## Enzyme kinetic assays

Furin inhibition constants of **M1-M5**, **F1**, and **F2** were determined at condition 1 (50 mM Hepes/NaOH, pH 7.4, 150 mM NaCl, 2 mM CaCl$_2$, 0.2% (v/v) Triton X-100 at 37 °C) with Pyr-Glu-Arg-Thr-Lys-Arg-7-amino-4-methylcoumarin (pERTKR-AMC, Bachem) as substrate. pH-dependent inhibition assays of **M1, M2, M5, M9**, and **M10** with furin were performed at condition 2 (50 mM Mes/NaOH, pH 6.0, 300 mM NaCl, 2 mM CaCl$_2$, 0.2% (v/v) Triton X-100 at 25 °C) or at condition 3 (50 mM Hepes/NaOH, pH 7.4, 300 mM NaCl, 2 mM CaCl$_2$, 0.2% (v/v) Triton X-100 at 25 °C) with Ac-RR(Tle)KR-AMC as substrate (substrate 10)[44]. **F1** and **F2** inhibition of furin T562R, PC7, and PC5 was investigated at condition 1 using the substrates pERTKR-AMC, H-RR(Tle)KR-AMC (substrate 11)[44] and pERTKR-AMC, respectively. For each measurement series, $K_m$-values were determined and used for calculation of the $K_i$-values. Concentrations of the enzymes and substrates used for $K_m$- and $K_i$-determinations are summarized in Table S3. The reactions were started by the addition of substrate, and the fluorescence was measured in a microplate reader (Spark with SparkControl v 3.2 or Infinite 200 with i-control 2.0.10.0, Tecan) at excitation and emission wavelengths of 380 nm and 460 nm, respectively. All measurement series were performed in triplicates ($n = 3$). The data were evaluated as replicate values in GraphPad Prism (version 10.2.2 GraphPad Software, La Jolla, CA). Enzyme kinetic data were evaluated using the prebuild "Michaelis-Menten" model to calculate $K_m$ and the Morrison equation for $K_i$ determination under tight-binding conditions. The enzyme concentration was fixed for curve fitting using the Morrison equation. For $K_i$ determinations of **F1** and **F2** with furin under tight-binding conditions, the enzyme concentrations were fitted, resulting in enzyme concentrations of 83 ± 7 pM and 97 ± 7 pM, respectively. The $K_i$-values of non-tight binding, competitive inhibition were evaluated using the model $v = V_{max}*[S]/([S] + K_m(1 + [I]/K_i))$ with $v$ = reaction velocity, $V_{max}$ = maximum velocity, $[S]$ = substrate concentration, $K_m$ = Michaelis-Menten constant, $K_i$ = inhibition constant and $[I]$ = inhibitor concentration. The standard errors of the curve fit are always given for the determined $K_i$ and $K_m$ values.

The inhibition mode of **F2** was investigated at condition 1 with 50 pM furin. IC$_{50}$ values were measured for inhibitor concentration ranges 6.3 pM–400.6 pM (at 25 µM and 50 µM pERTKR-AMC) and 12.5 pM–801.2 pM (at 100 µM, 150 µM, and 200 µM pERTKR-AMC). For all substrate concentrations measurements without inhibitor were performed as well. All measurements were performed in triplicates. IC$_{50}$ values were determined in GraphPad Prism fixing the upper and lower limits to the reaction velocity without inhibitor and 0, respectively.

## nanoDSF measurements

The melting temperatures ($T_m$) of the isolated PC1/3-prodomain were determined at 50 mM $NaH_2PO_4$/NaOH, pH 6.0, 300 mM NaCl and 50 mM $NaH_2PO_4$/NaOH, pH 7.4, 300 mM NaCl using a Tycho nanoDSF instrument (NanoTemper, Germany) using ~20 µM protein. $T_m$-values of PC1/3-prodomain-furin complexes and isolated furin were determined in 50 mM Mes/NaOH, pH 6.0, 150 mM NaCl, 2 mM $CaCl_2$, and 50 mM Hepes/NaOH, pH 7.4, 150 mM NaCl, 2 mM $CaCl_2$ using 3.8 µM furin and 5.8 µM prodomain. To ensure saturation of furin with the prodomain, a molar ratio of 1:1.5 (furin:PC1/3pro) was applied. Melting curves were obtained from the ratio between the fluorescence intensities at 330 and 350 nm. Melting curves were evaluated in GraphPad Prism determining $T_m$ from the peak of the first order derivatives. All measurements were performed in triplicates and the errors are given as standard deviations.

## Protein crystallography

The **M2**:furin complex was prepared mixing 4 mg/ml furin, 7.1 mg/ml **M2** and 10 mM Hepes/NaOH, pH 7.4, 2 mM $CaCl_2$ in a ratio (v/v) of 7.5:1:2 to obtain a molar ration of 1.3/1 of **M2**/furin and a final buffer concentration of 10 mM Hepes/NaOH, pH 7.4, 100 mM NaCl, 2 mM $CaCl_2$. The protein mixture was concentrated to 15 mg/ml using an ultrafiltration membrane with 30 kDa cutoff and initially crystallized in a sitting drop vapor diffusion setup mixing protein solution and 2.8 M sodium acetate pH 7.0 in a ratio 1:1. The very thin plate-like crystals appeared after 8 months. To reproduce the crystals, the **M2**:furin mixture was concentrated to 27.0 mg/ml and mixed with 3.75 M sodium acetate pH 7.0, 10 mM Hepes/NaOH pH 7.4, 1 mM $CaCl_2$ in a ratio 1:1 in sitting drop crystallization setups. The dops were incubated overnight and seeded from the initial crystallization condition. Very small crystals (spacegroup P1, longest dimension of ≤20 µm) were reproduced within several days. For data collection crystals were mounted on a micro-mesh (20 µm grid size, MiTeGen) draining excessive reservoir solution with a small piece of chromatography paper (Whatman), and flash-cooled in $N_2$-l. Diffracting crystals were identified using a mesh scan and measured with the 10 µM aperture at ID23-1 at the European Synchrotron Radiation Facility (ESRF)[45], using MxCube3 (https://www.mxcube.org). The data were processed using XDS[46] (v Jan 10, 2022) with XDS-APP[47] (v 2.9) and programs of the CCP4 program suite[48] (v 7.0.078). 80°-datasets of 19 different crystals were evaluated for data merging using a custom python script[49] (v1). The script iteratively tested all datasets for successive merging with the other datasets in XSCALE[46] (v Jan 10, 2022) to reach the highest possible completeness while obtaining highest possible I/sig- and $CC_{1/2}$-values of the merged datasets. Data merging of the clustering datasets from this analysis was manually optimized (Table S1).

Crystals of isolated **M2** grew in several crystallization conditions of the initial screen of the **M2**:furin complex (see above). Best diffracting crystals grew in 0.2 M sodium citrate tribasic dihydrate, 20% (w/v) Polyethylene glycol 3,350. Crystals were flash-cooled in $N_2$-l and data were collected at ID23-1 (ESRF) using MxCube3 (https://www.mxcube.org). Diffraction data were processed using XDS[46] (v Jun 30, 2023) with XDS-APP[47] (v2.9) and programs of the CCP4 program suite[48] (v 7.0.078).

**M5**:furin complex was prepared mixing 3.8 mg/ml furin, 8.1 mg/ml His$_6$-tagged **M5** in a ratio (v/v) of 8.3:1 to obtain a molar ratio of 1.4/1 of **M5**/furin. Water and 100 mM $CaCl_2$ were added to obtain a final buffer concentration of 8.2 mM Hepes/NaOH, pH 7.4, 100 mM NaCl, 2 mM $CaCl_2$. The protein solution was concentrated and prepared for crystallization as described for **M2**:furin (see above), obtaining heavily intergrown crystals in 100 mM sodium citrate, pH 5.9, 0.5 mM $(NH_4)_2SO_4$, and 1.0 M $LiSO_4$. The crystals were suspended in 100 mM sodium citrate, pH 5.9, 0.5 mM $(NH_4)_2SO_4$, and 1.38 M $LiSO_4$, and 1 mM $CaCl_2$, vortexed with glass beads (diameter 0.5 mm) and centrifuged @ 3000 × $g$. Aliquots of the supernatant were frozen in $N_2$-l, stored @

−80 °C, and used directly after thawing for micro-seeding. Single crystals of **M5**:furin were obtained mixing the protein mixture in a 1:1 ratio with 100 mM sodium citrate, pH 5.9, 0.5 mM $(NH_4)_2SO_4$, and 0.8 M $LiSO_4$ followed by an overnight incubation and micro-seeding. Needle-shaped micro-crystals (spacegroup $P2_1$, smallest dimension <5 µm, longest dimension <20 µm) were observed within several days. For cryo-protection a micro-mesh (400 × 400 µm, 10 µm grid size, MiTeGen) filled with crystals was transferred to 100 mM sodium citrate, pH 5.9, 0.5 mM $(NH_4)_2SO_4$ and 1.38 M $LiSO_4$ and 1 mM $CaCl_2$ supplemented with 20% (v/v) glycerol, immediately drained on a small piece of chromatography paper (Whatman) and flash-cooled in $N_2$-l. Using a mesh-and-collect data collection strategy at the microfocus beamline ID23-2 at the ESRF[50], we identified diffracting crystals and collected 94 10°- or 20°-datasets. Data collection software was MxCube3 (https://www.mxcube.org). The datasets were automatically processed with XDS[46] (v Feb 5, 2021) in combination with XDS-APP[47] (v 2.9) or alternatively by running XDS in a custom TCL script[49] (v1). Using this custom script, refinement of the detector distance and unit cell constants was done in consecutive integration runs, which resulted for many datasets in better data quality. The data of 79 microcrystals were merged using our custom Python script as described above with XSCALE[46] (v Jan 10, 2022; Table S1).

The crystal structure of **M2** was solved by molecular replacement (MR) using the PC1/3-prodomain structure from the AlphaFold Protein Structure Database (https://alphafold.ebi.ac.uk/)[51,52] as search model in PHASER[53] (v 2.8.3). In contrast, the NMR-structure of the mouse PC1/3-prodomain (PDB-ID 1KN6)[54] did not reveal a useful MR-solution in our hands, probably because of huge structural differences to the structure of the human PC1/3-prodomain of this study (Cα-RMSD = 2.9 Å, Fig. S18). The structures of **M2**:furin and **M5**:furin were solved by molecular replacement (MR) using the PDB-ID 5JXG[5] and the herein determined structure of isolated **M2** (see below) as search models in PHASER[53] (v 2.8.3). COOT[55] (v 0.8.8) was used for model building. Refinement and the calculation of composite omit maps were performed in PHENIX[56] (v1.20.1). For the refinement of **M5**:furin automatic ncs-group assignment was enabled. Electron density composite omit maps were calculated in PHENIX[56] (v 1.20.1). PYMOL (v 2.0.7) was used for molecular graphics (http://www.pymol.org) and structural alignments.

For analysis of the protein content in **M5**:furin crystals, 5 µl microcrystal suspension was harvested from the crystallization screen. The crystals were centrifuged 1 min at 17000 × $g$, and the mother liquor was removed. The pellet was washed three times by addition of 50 µl 100 mM sodium citrate, pH 5.9, 500 mM $(NH_4)_2SO_4$, 1.38 M $LiSO_4$, and 1 mM $CaCl_2$ and centrifugation at 3000 × $g$. 25% of the remaining microcrystal pellet was loaded to SDS-PAGE together with furin (2.5 µg) and His$_6$-tagged **M5** (1 µg) as controls.

## NMR spectroscopy

NMR spectra were recorded on a 600 MHz Avance III-HD spectrometer (Bruker Biospin) equipped with a QXI ($^1$H/$^{13}$C/$^{15}$N/$^{31}$P) probe at 298 K unless stated otherwise. Samples were measured in standard 5 mm tubes (Armar, TA quality) containing 0.2 mM protein, 50 mM $NaH_2PO_4$/$Na_2HPO_4$, 150 mM, pH 7.4, 7% (v/v) $D_2O$, and 93% ddH₂O. Sequence specific assignment was achieved using 2D $^1$H-$^{15}$N HSQC, 2D $^1$H-$^{13}$C HSQC, 3D HNCA, 3D CBCA(CO)NH, 3D HNCACB, 3D HNCO, 3D HN(CA)CO, 3D (H)CCH-TOCSY, 3D $^{15}$N-edited NOESY-HSQC, 3D $^{13}$C-edited NOESY-HSQC spectra[57]. For the 3D (H)CCH-TOCSY spectrum a mixing time of 21.7 ms was used. All NOESY experiments were recorded using a mixing time of 120 ms unless stated otherwise. A high resolution 2D $^1$H-$^1$H NOESY with $^{15}$N decoupling and a 2D $^1$H-$^1$H TOCSY spectrum with $^{15}$N decoupling were recorded using the $^{15}$N sample. Steady state $^{15}$N{$^1$H}heteronuclear Overhauser effects were measured according to Farrow et al.[58]. For the assignment of the tautomeric forms of imidazole rings of histidines 2D $^1$H-$^{15}$N HMBC spectra were

used[59]. Additional experimental details of the applied NMR experiments are provided in Table S4.

For estimating pKa values, we measured 2D $^1$H-$^{15}$N HMBC spectra at pH values of 7.4, 6.8, 6.6, 6.3, and 6.0 at 278 K. Titration of the sample by direct addition of acid was not possible and resulted in precipitation of the protein. Therefore, the protein was re-buffered gently to 50 mM $NaH_2PO_4$/$Na_2HPO_4$, 150 mM NaCl with the adjusted pH at 80 μM concentration by dialysis at 4 °C. Subsequently, the protein was concentrated to 0.2 mM using ultrafiltration.

Data were processed using Topspin 3.5/3.6 (Bruker Biospin) and analyzed with Sparky 3.114 (T. D. Goddard and D. G. Kneller, SPARKY 3, University of California, San Francisco, USA). Spectra were referenced to 2,2-dimethyl-2-silapentane-5-sulfonic acid (DSS) using an external sample of 2 mM sucrose and 0.5 mM DSS in 10% $D_2O$/90% $H_2O$ (Bruker Biospin) for any applied temperature. The chemical shifts of $^{13}$C and $^{15}$N were referenced using the IUPAC-IUB recommended chemical shift referencing ratios Ξ of 0.251449530 ($^{13}$C) and 0.10132918 ($^{15}$N).

## Alphafold calculations
The structure of **F1** in complex with furin has been predicted using the Alphafold 3 server (https://alphafoldserver.com[60]). The input was defined using the structured part of human furin as observed for PDB ID 5JXG[5], the complete **F1** sequence, three $Ca^{2+}$-ions, and one $Na^+$-ion. The confidence score (pLDDT) mapped on the structure of the predicted model is shown in Fig. S15.

## H7N7/SC35M infection and inhibition of multicycle virus replication
Recombinant influenza A virus H7N7/SC35M (reverse genetics system was kindly provided by Jürgen Stech (Friedrich Loeffler Institute, Greifswald - Isle of Riems, Germany)). H7N7/SC35M was propagated in Madin-Darby canine kidney II (MDCKII) cells (Source: Institute of Virology, Philipps University Marburg; not authenticated as part of this study) in infection medium (Dulbecco's Modified Eagle Medium (DMEM, Gibco, Thermo Fisher Scientific, Paisley, UK) supplemented with 0.1% bovine serum albumin (SIGMA-ALDRICH Co.), penicillin, streptomycin (Gibco, Thermo Fisher Scientific), and 2 mM L-glutamine (Gibco, Thermo Fisher Scientific). Virus-containing cell supernatants were cleared from cell debris by low-speed centrifugation and stored as virus stocks at −80 °C. For inhibition of H7N7/SC35M multicycle replication, human lung adenocarcinoma cells (A549; Source: Institute of Virology, Philipps University Marburg; not authenticated as part of this study) were grown in DMEM supplemented with 10% fetal bovine serum (FBS, Gibco, Thermo Fisher Scientific), penicillin, streptomycin and 2 mM L-glutamine until 95 % confluence. Growth medium was removed, cells were washed with PBS def. and infected with H7N7/SC35M at a low multiplicity of infection (MOI) of 0.0001 for 1 h at 37 °C, 5% $CO_2$. Inoculum was removed, cells were washed with PBS def. and fresh medium with or without inhibitors was added. Cells were then further incubated at 37 °C, 5% $CO_2$ for 72 h in total. 50 μL cell supernatant was collected at different time points and stored at −20 °C until plaque assay titration.

## Plaque assay titration SC35M
MDCKII cells were cultured until 95% cell confluence was reached, growth medium was removed, and cells washed with PBS def. Cells were then inoculated with 10-fold serial dilutions of virus-containing cell supernatants in DMEM supplemented with 0.1% BSA, penicillin, streptomycin, and 2 mM L-glutamine and incubated at 37 °C, 5% $CO_2$ for 1 h. Inoculum was removed and replaced with 50% v/v double-concentrated MEM (2xMEM (Gibco) supplemented with 0.6% BSA, penicillin, streptomycin, 4 mM L-glutamine, and 50% cellulose solution (Avicel, Sigma Aldrich) 2.5% w/v in double-distilled $H_2O$) and incubated for 48 h at 37 °C, 5% $CO_2$. Overlay was removed and cells were washed three times with PBS def. Cells were then fixed with 4% w/v

formaldehyde in PBS def. at 4 °C for 30 min. Next, cells were stained with Giemsa's Azur eosin methylene blue solution (Merck Millipore). Virus-infected areas were counted, and plaque-forming units per mL were calculated.

## Analyses of HA cleavage
A549 cells were cultured in 12-well plates and grown to 95% confluence in DMEM supplemented with 10% fetal bovine serum (FBS), penicillin, streptomycin, and 2 mM L-glutamine at 37 °C, 5% $CO_2$. Growth medium was removed, cells were washed with PBS def. and infected with H7N7/SC35M at a high MOI of 0.1 in DMEM supplemented with 0.1% BSA, penicillin, streptomycin, and 2 mM L-glutamine for 1 h at 37 °C, 5% $CO_2$. Inoculum was removed, cells were washed with PBS def., and then treated with inhibitors at the indicated concentrations and incubated in the presence of inhibitors for 24 h at 37 °C, 5% $CO_2$. Subsequently, cells were washed with PBS def., lysed in Cell Lysis buffer (CellLytic M, Sigma-Aldrich Pty Ltd, Merck KGaA) supplemented with 0.1% v/v Protease Inhibitor-Cocktail (Protease Inhibitor-Cocktail derived from bovine lung, Sigma-Aldrich Pty Ltd, Merck KGaA), mixed with 6x reducing Laemmli buffer, and heated at 95 °C for 10 min. Cell lysates were subjected to SDS-PAGE (12% polyacrylamide gel), transferred to a polyvinylidene difluoride (PVDF) membrane (GE Healthcare, Freiburg, Germany), and detected with primary antibodies (polyclonal rabbit serum against H7, diluted 1:500, Sino Biological Inc., Catalog-No.: 40104-T62 and monoclonal Anti-α-Tubulin antibody produced in mouse, diluted 1:1000, SIGMA-ALDRICH Co., Catalog-No.: 40104-T62) and peroxidase-conjugated secondary anti-rabbit (diluted 1:6000, Agilent Dako, Catalog-No.: P021702-2) and anti-mouse antibodies (diluted 1:6000, Agilent Dako, Catalog-No.: P026002-2). Proteins were visualized using the ChemiDoc XRS+ system with Image Lab software (Bio-Rad).

## Stability assays in cell culture
A549 cells were cultured in 24-well plates in DMEM supplemented with 10% fetal bovine serum (FBS), penicillin, streptomycin, and 2 mM L-glutamine at 37 °C, 5% $CO_2$ until 95% cell confluence was reached. Growth medium was removed, cells washed with PBS def., and DMEM supplemented with 0.1% BSA, penicillin, streptomycin, and 2 mM L-glutamine was added. One micrometer of PC1/3-prodomain variants and PC1/3-prodomain-nanobody fusion proteins were added to the medium and cultivated 72 h. As negative control buffer without protein was added to the cells. Medium samples were taken at 0 h (directly after addition of the proteins), 24 h, or 72 h and stored at −20 °C until enzyme activity measurement. As negative control buffer without protein was added to the cells. Activity of furin was measured at 25 °C in cell culture medium diluted 1:20 in 45 mM Hepes/NaOH, pH 7.4, 135 mM NaCl, 1.8 mM CaCl2, 0.18% Triton X-100, 20 μM pERTKR-AMC, and 2 nM furin using a microplate reader (Infinite 200 with i-control 2.0.10.0, Tecan). The activity of furin in presence of fresh cell culture medium was set to 100% activity and used to normalize the activities of conditioned cell culture medium. Inhibition by conditioned medium was calculated: 100%−activity [%]. All measurements were performed in triplicate ($n = 3$). For western blot analysis 3 μl of conditioned cell culture medium containing **F1** or **F2** was subjected to SDS-PAGE, transferred to a nitrocellulose membrane (GE Healthcare, Freiburg, Germany), and detected with a primary anti-nanobody antibody (rabbit anti-camelid VHH, mAb cocktail, diluted 1:2000, GenScript, Catalog-No.: A02014) and an anti-rabbit peroxidase-conjugated secondary antibody (diluted 1:40000, Cell Signaling Technology, Catalog-No.: 7074S). Proteins were visualized using the ChemiDoc XRS+ system (Bio-Rad) with Image Lab software (v 6.0.1, Bio-Rad).

## Reporting summary
Further information on research design is available in the Nature Portfolio Reporting Summary linked to this article.

## Data availability

Structures of **M5**:furin, **M2**:furin and **M2** are deposited in the protein database (PDB) under the PDB-IDs 9FID, 9FIE, and 9FIC. The chemical shift assignments are made publicly available at the BioMagResBank[61] under the accession code 52871. The model of the **F1**:furin complex predicted with Alphafold 3, all input data used, and the complete output data are available at https://www.modelarchive.org under https://doi.org/10.5452/ma-mg3kp. PDB-IDs cited in this study: 1KN6, 5JXG. Source data are provided with this paper as a Source Data File.

## Code availability

The code of the scripts used in this work are available under https://doi.org/10.5281/zenodo.16356790.

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

## Acknowledgements

We acknowledge the European Synchrotron Radiation Facility (ESRF) for provision of synchrotron radiation at the beamlines ID23-1 and ID23-2 as part of the Austrian BAG proposal and thank the scientific staff for assistance. Funding was provided by the Austrian Science Fund (FWF) to S.O.D. (P36648-B), The State of Hesse LOEWE Center DRUID (project D1) to E.B.-F., the Erasmus+ program of the European Union to S.M. Open access publication was supported by the Paris Lodron University of Salzburg Publication Fund. We thank Torsten Steinmetzer (Philipps University Marburg) for providing the substrates 10 and 11.

## Author contributions

R.K., K.B., M.S., E.B.-F., H.B., and S.O.D. designed research; R.K., K.B., L.S.E., S.M., M.S. and S.O.D. performed research; R.K., K.B., L.S.E., S.M., M.S., E.B.-F., H.B., and S.O.D. analyzed data; R.K., K.B., and S.O.D. wrote the paper. All authors reviewed and revised the paper.

## Competing interests

The authors declare no competing interests.
