## [Transparent Peer Review file · Nature Communications]

Structural insights into proprotein convertase activation facilitate the engineering of highly specific furin inhibitors.

Corresponding Author: Dr Sven Dahms

Version 0:

Reviewer comments:

Reviewer #1

(Remarks to the Author)

The manuscript by Klaushofer and colleagues from the Brandstetter group reports on the notable achievement of highly specific inhibitors of the proprotein convertase (PC) furin. This is a subtilisin-like serine peptidase that plays key roles in many physiological processes, as well as in disease, as shown for the entry mechanism of SARS-CoV-2 into its target cells. As happens with other families of paralogous peptidases, e.g. matrix metallopeptidases, PCs possess highly similar active sites, which poses an extraordinary difficulty to developing highly selective inhibitors for one in particular of the nine mammalian PCs known.

The authors took advantage of the prodomain (PD), an N-terminal well-folded ~80-residue moiety that acts as a chaperone for correct folding and a latency maintainer in PCs and other subtilisins. The PD of PC1 served as a scaffold for designing a highly selective and potent inhibitor of furin, which was attached to the C-terminus of the furin-binding Nb14 nanobody, which binds to furin to an active-site distal surface patch. This resulted in a picomolar inhibitor of furin that effectively suppressed the replication of a furin-dependent H7N7-influenza virus in a cell-based assay.

This work, which paves the way for the development of specific inhibitors targeting other PCs and peptidases in general, is of remarkable significance, and I wholeheartedly endorse its publication in Nature Communications.

Minor points:

- In the abstract, please state that there are nine mammalian PCs, among which PC1 and furin. Currently, the first sentence of the second paragraph is confusing.
- Legend to Figure 3: I only see one green and one blue arrowhead.
- Line 147 should start with a capital letter.
- Line 151: What is a "micro-crystallography approach" ?

Reviewer #2

(Remarks to the Author)

The proprotein convertase furin is an important biological target not only in the case of coronavirus infection but also for a variety of different viral and bacterial pathogens. This study presents an innovative strategy for furin inhibition that has resulted in the most potent inhibitors known to date. The authors have here combined a PC1/3 propeptide domain engineered to obtain maximal furin inhibition with a single chain antibody that directs inhibitory activity to furin. They provide extensive biophysical and crystallographic evidence as to binding mechanism; describe stability and kinetic assays demonstrating specificity; and finally use a biological assay to demonstrate inhibitor potency against viral replication as well as against HA cleavage in culture. Of note, the specificity for furin for this novel inhibitor relative to related convertases is remarkably high, suggesting reduced off-target effects.

The most noteworthy accomplishment of this study is clearly the successful engineering of the PC1/3 propeptide-Nb14

nanobody fusion protein to produce a specific and potent furin inhibitor- picomolar in vitro, and micromolar inhibition in cell culture. The beauty of this construct is that it contains a potent inhibitory portion (the engineered propeptide) along with a targeting portion that confers specificity to the inhibitor and also assists in inhibition (the camelid nanobody, which has previously been shown to block the entry of large substrates to the furin active site). This novel strategy could be conceivably adapted to specifically inhibit other convertases. The authors also discuss the potential use of these constructs as cellular furin sensors, an interesting and highly viable approach to measuring cellular furin activity. More speculatively- and not discussed here - this inhibitor could also conceivably also be genetically encoded in a virus for controlled expression in a therapeutic or experimental setting.

Comments on improving the manuscript include the following:

1. Summarizing statements throughout the crystallography section (lines 145-212) would improve the accessibility of the manuscript for the general reader.
2. The mutational analysis which follows line 212 tests a prior hypothesis on PC1/3 propeptide maturation and pH sensitivity - but uses furin rather than PC1/3 to cleave the PC1/3 propeptide. The logic of testing these different mutations does not immediately relate to the objective of the paper, ie creating a potent furin inhibitor. This discussion could easily be restructured/re-titled for clarity, ie "Engineering of a stable inhibitor:enzyme complex."
3. The "11 residue" linker sequence for the fusion protein (line 280) is listed as only 10 residues in the Methods (line 465), suggesting a missing residue.
4. The Figure 5 legend needs to contain error bar definition and replicate number.
5. Line 350- Discussion- the statement that prodomain proteolysis is "initiated by a drop in pH during secretion of the proteases" is incorrect. "Trafficking" would be the correct word. Prodomain cleavage of both PC1/3 and furin (and most other convertases) occurs well before secretion, ie during trafficking through the secretory pathway.
6. Line 362 It isn't clear to readers naïve to proprotein convertase biochemistry why the pH change is relevant to trafficking and/ or activation. A little more background re secretory biology could help with this.

Minor comments:

- The term PC1 has been replaced in the field by PC1/3 to reflect both the co-discoverers of this enzyme as well as to avoid confusion with the abbreviation for "principal component".
- Motive and motif are confused throughout the ms.
- Reference 30 is incorrect; the authors mean to cite Ref 31.
- Defining M and F abbreviations in the figure legends would help general readability.

Reviewer #3

(Remarks to the Author)

In this manuscript, Klaushofer et al report atomic level structural and functional insights into the basis of proprotein convertase (PC) activation, and report the design and efficacy of a potent new mutant PC1:nanobody fusion protein specific for furin. The results are directly relevant to the challenges of designing specific inhibitors to the broad family of PCs whose dysregulation in vivo is associated with a wide range of diseases. The manuscript is well-written and the results are presented in a logical and descriptive manner.

I feel the manuscript requires the following minor revisions prior to acceptance for publication.

1. Histidine 72:

The authors provide solid X-ray structure and NMR NOE evidence demonstrating the critical role of the side chain of His72 in PC1 which mediates a small hydrogen bond network with nearby residues.

Some questions here:

- i. Clearly H72 adopts a Ne2H neutral tautomer in solution at pH 7.4 based upon the structure, and this must be definitive in the 2D 1H-15N HMBC. This is a critical point. Can the authors please add a supplemental figure showing their 2D 1H-15N HMBC and the distinctive pattern expected for this tautomer, and also those of the other histidines in this PC1 domain. His75 and His 85 are also nearby in this important secondary cleavage site loop.
- ii. The 13C15N-labeled PC1 sample affords an excellent opportunity to explore the pKa's of the histidines in this domain, via the pH dependence of the 1H/13C chemical shift of the side chain ϵ 1's, which are assigned. Can the authors add this experiment to the Supplemental of this study, given the importance of H72 and the critical role pH plays in the proteolytic processing of PC's.

2. NMR assignments, especially of the aromatic residues.

- i. With the BMRB deposition on hold, please report in the Supplementary your statistics for the percent backbone and overall assignment of the PC1 domain
- ii. In addition to the histidines, there are several important NOEs to aromatics reported. How were the aromatic side chains assigned? I am guessing a combination of the Yamazaki et al (1993) experiments ($^{13}\text{C}\beta \rightarrow ^1\text{H}\delta/\text{H}\epsilon$) and ^{13}C -NOESY, and the $\text{H}\delta$ 2's and $\text{H}\epsilon$ 1's of the histidines assignments fall out of the 2D 1H-15N HMBC. Please describe and cite the methods used.

3. Secondary structure comparison of PC1 from X-ray and NMR (Fig. S8).

- i. In the chemical shift indexing (CSI) plot caption please specify for the reader that the rc subscript in the y-axis represents the diamagnetic random coil chemical shift
- ii. Although +/- 1 residue differences in the exact termini of secondary structural elements are often observed between X-ray and NMR structural data of the same protein, I think the 75-80 region of the sequence warrants some further experimentation since there is a small strand observed in the Xray structure but "not quite" a strand (gray) in the CSI data. This may reflect

some interesting dynamics in this important stretch which is juxtaposed to the critical H72 residue, and contains the secondary cleavage site.

I suggest the authors append residue specific ^{15}N dynamical data in the Fig. S8 plot. i.e., S2 order parameters from ^{15}N T1, T2 and $^{15}\text{N}\{^1\text{H}\}$ heteronuclear NOE data or at the very least the het-NOE.

4. Linker in the fusion protein:

How did the authors design the linker between PC1 and nb14 in the fusion protein? Is it vulnerable to degradation in vivo and, if so, what can be done to mitigate this without disturbing the efficacy of the fusion protein toward furin inhibition?

5. Minor experimental question: in the main text Methods the authors report using an NOE mixing time of 120ms, but it is 100ms in the Supplemental Information (caption to S13)

6. Typos:

i. Fig. 2 caption: Change Asn72 to Asn74 in the text

ii. Labels in Fig. S13 NOESY spectra: please use symbols (Greek letters) where appropriate:

i.e. He1 \rightarrow He ϵ 1; also, β , γ , δ

Reviewer #4

(Remarks to the Author)

In this manuscript, Klaushofer et al. structurally investigated the interactions between PC1-25 prodomain and furin, and engineered prodomain-based inhibitors against furin by mutating different amino acid residues near the cleavage site. The activity of the inhibitor was significantly increased by fusion expression with a furin-specific nanobody. Overall, this work deepens our understanding of the mechanism of furin activation and has potential applications.

In terms of inhibiting viral replication, F1 is much better than F2 (Figure 4), and the authors claim that it is largely due to the structural stability of the F1 prodomain [as stated in lines 387-389: "we observed a higher stability of the prodomain part of F1 (M9) in cell culture compared to the prodomain part of F2 (M5)"]; however, a direct comparison of F1 and F2 showed no significant difference in stability in cell culture (Figure 5). Given that the function of furin is performed in the trans-Golgi network (TGN), it is advisable to assess the differences in the intracellular environment: how does F1 reach the site of action? Or what is the half-life of F1 in a cell?

Here are some other minor suggestions for improving the manuscript:

Line 279 "the C-terminus auf the nanobody" should be "the C-terminus of the nanobody"

Line 315 "with a 10.000-fold reduction" should be "with a 10,000-fold reduction"

Lines 579-582 "probably because of huge structural differences to the structure of the human PC1-prodomain of this study ($\text{C}\alpha\text{-RMSD} = 2.9 \text{ \AA}$)": a supplementary figure could be helpful for understanding the "huge structural differences".

Version 1:

Reviewer comments:

Reviewer #1

(Remarks to the Author)

The authors addressed all my concerns from the first round of review in a satisfactory manner. The manuscript should be ready for acceptance now.

Reviewer #2

(Remarks to the Author)

The authors have been remarkably responsive to the comments of all of the reviewers, thus improving the accessibility and the precision of the language.

Reviewer #3

(Remarks to the Author)

In their revised manuscript, Klaushofer et al made an excellent effort at addressing the questions raised in the first submission, most notably performing a number of additional NMR experiments. The new 2D ^{15}N , ^1H -HMBC experiment in particular is a beautiful result and conclusively reveals the neutral tautomeric state of the important histidines in the key hydrogen bonding network presented in this paper. Although using the $^{15}\text{N}\delta_1$ chemical shift to determine pKa's provides less of a dynamic range than, for example, the $^{13}\text{C}\epsilon_1$ chemical shift, the point is clearly made that the pKa's of H72, H75 and H85 are significantly lower than normal.

One small nomenclature correction to the revisions: for consistency, in line 246 of the revised main text, please change:

(....He2 tautomer) to (....Ne2 tautomer)

Reviewer #4

(Remarks to the Author)

The authors have adequately addressed the issues I raised. I support the publication of the revised paper.

RESPONSE TO REVIEWERS

Structural insights into proprotein convertase activation facilitate the engineering of highly specific furin inhibitors.

Rupert Klaushofer^{1,2‡}, Konstantin Bloch^{3‡}, Luisa Susanna Eder^{1,2}, Simone Marzaro¹, Mario Schubert¹, Eva Böttcher-Friebertshäuser³, Hans Brandstetter^{1,2} and Sven O. Dahms^{1,2,*}

¹ Department of Biosciences and Medical Biology, University of Salzburg, Hellbrunner Straße 34, A-5020 Salzburg, Austria

² Center for Tumor Biology and Immunology (CTBI), University of Salzburg, Hellbrunner Straße 34, A-5020 Salzburg, Austria

³Institute of Virology, Philipps University, Hans-Meerwein-Str. 2, D-35043 Marburg, Germany

[‡] Authors contributed equally

* To whom correspondence should be addressed:

Sven O. Dahms: sven.dahms@plus.ac.at, Tel: +43-662-80447277

Our responses to the reviewers' questions are highlighted in red.

Reviewer #1 (Remarks to the Author):

The manuscript by Klaushofer and colleagues from the Brandstetter group reports on the notable achievement of highly specific inhibitors of the proprotein convertase (PC) furin. This is a subtilisin-like serine peptidase that plays key roles in many physiological processes, as well as in disease, as shown for the entry mechanism of SARS-CoV-2 into its target cells. As happens with other families of paralogous peptidases, e.g. matrix metallopeptidases, PCs possess highly similar active sites, which poses an extraordinary difficulty to developing highly selective inhibitors for one in particular of the nine mammalian PCs known.

The authors took advantage of the prodomain (PD), an N-terminal well-folded ~80-residue moiety that acts as a chaperone for correct folding and a latency maintainer in PCs and other subtilisins. The PD of PC1 served as a scaffold for designing a highly selective and potent inhibitor of furin, which was attached to the C-terminus of the furin-binding Nb14 nanobody, which binds to furin to an active-site distal surface patch. This resulted in a picomolar inhibitor of furin that effectively suppressed the replication of a furin-dependent H7N7-influenza virus in a cell-based assay.

This work, which paves the way for the development of specific inhibitors targeting other PCs and peptidases in general, is of remarkable significance, and I wholeheartedly endorse its publication in Nature Communications.

Minor points:

-In the abstract, please state that there are nine mammalian PCs, among which PC1 and furin. Currently, the first sentence of the second paragraph is confusing.

>>> We thank the reviewer for the helpful comments. According to the reviewers' suggestion we included this statement in the first sentence of the abstract of the revised manuscript: "Proprotein convertases (PCs), including furin and PC1/3 among nine mammalian homologues, mediate the maturation of numerous secreted substrates by proteolytic cleavage.". To keep the word limit of 200 words, we optimized the wording of the abstract.

-Legend to Figure 3: I only see one green and one blue arrowhead.

>>> We apologize for the misleading coloring scheme and replaced the arrowheads with direct labels of the respective termini in figure 3 of the revised manuscript. The figure legend of the revised manuscript was adapted accordingly.

-Line 147 should start with a capital letter.

>>> We have adapted this issue in the revised manuscript. This sentence reads now: "The pH-dependent destabilization ..."

-Line 151: What is a "micro-crystallography approach" ?

>>> In the revised manuscript we give a more comprehensive explanation of this fact: "These crystals were too small to obtain complete datasets of good quality from single crystals. Thus, we applied a micro-crystallography approach for data collection and measured partial datasets from many small crystals that were merged afterwards to obtain a complete dataset."

Reviewer #2 (Remarks to the Author):

The proprotein convertase furin is an important biological target not only in the case of coronavirus infection but also for a variety of different viral and bacterial pathogens. This study presents an innovative strategy for furin inhibition that has resulted in the most potent inhibitors known to date. The authors have here combined a PC1/3 propeptide domain engineered to obtain maximal furin inhibition with a single chain antibody that directs inhibitory activity to furin. They provide extensive biophysical and crystallographic evidence as to binding mechanism; describe stability and kinetic assays demonstrating specificity; and finally use a biological assay to demonstrate inhibitor potency against viral replication as well as against HA cleavage in culture. Of note, the specificity for furin for this novel inhibitor relative to related convertases is remarkably high, suggesting reduced off-target effects.

The most noteworthy accomplishment of this study is clearly the successful engineering of the PC1/3 propeptide-Nb14 nanobody fusion protein to produce a specific and potent furin inhibitor- picomolar in vitro, and micromolar inhibition in cell culture. The beauty of this construct is that it contains a potent inhibitory portion (the engineered propeptide) along with a targeting portion that confers specificity to the inhibitor and also assists in inhibition (the camelid nanobody, which has previously been shown to block the entry of large substrates to the furin active site). This novel strategy could be conceivably adapted to specifically inhibit other convertases. The authors also discuss the potential use of these constructs as cellular furin sensors, an interesting and highly viable approach to measuring cellular furin activity. More speculatively- and not discussed here - this inhibitor could also conceivably also be genetically encoded in a virus for controlled expression in a therapeutic or experimental setting.

Comments on improving the manuscript include the following:

1. Summarizing statements throughout the crystallography section (lines 145-212) would improve the accessibility of the manuscript for the general reader.

>>> We thank reviewer #2 for positive feedback. To improve the readability of the crystallography section we included two summarizing statements in the crystallography section. Lines 186-188 of the revised manuscript reads "In conclusion, the binding of the PC1-prodomain to furin is mediated by exosite (globular part) as well as substrate-like interactions (C-terminus). The structural data suggests a specific destabilization of the secondary cleavage site loop at acidic pH." Lines 224-225 of the revised manuscript reads "Based on these findings we conclude that the protonation of His72 would prevent its sidechain-mainchain interactions and destabilize the conformation of the secondary cleavage site loop."

2. The mutational analysis which follows line 212 tests a prior hypothesis on PC1/3 propeptide maturation and pH sensitivity - but uses furin rather than PC1/3 to cleave the PC1/3 propeptide. The logic of testing these different mutations does not immediately relate to the objective of the paper, ie creating a potent furin inhibitor. This discussion could easily be restructured/retitled for clarity, ie "Engineering of a stable inhibitor:enzyme complex."

>>> According to the reviewers' suggestion we included a headline of this passage at line 218 of the revised manuscript: "Engineering of a stable inhibitor:enzyme complex by exchange of His72."

3. The "11 residue" linker sequence for the fusion protein (line 280) is listed as only 10 residues in the Methods (line 465), suggesting a missing residue.

>>> In the revised manuscript, we included a statement about the linker between nanobody and prodomain in the Methods: “ The PC1-prodomain was fused C-terminal to the nanobody using the linker sequence GQVTSAGASGQ (Table 3).”.

4. The Figure 5 legend needs to contain error bar definition and replicate number.

>>> As suggested by reviewer #2 we included a statement in the legend of figure 5 of the revised manuscript: “The experiment was performed in triplicates (Mean values and standard deviations are shown; open circles represent individual replicates).” We also included the values of the individual replicates in figure 3 of the revised manuscript.

We also checked other figures for similar issues and added more comprehensive statements about the number of replicates measured and errors in the figure legends when appropriate. In addition, we added more comprehensive statements about the number of replicates measured and errors in the Methods section when appropriate.

5. Line 350- Discussion- the statement that prodomain proteolysis is “initiated by a drop in pH during secretion of the proteases” is incorrect. “Trafficking” would be the correct word. Prodomain cleavage of both PC1/3 and furin (and most other convertases) occurs well before secretion, ie during trafficking through the secretory pathway.

>>> We changed the wording in the revised manuscript according to the reviewer’s suggestions.

6. Line 362 It isn’t clear to readers naïve to proprotein convertase biochemistry why the pH change is relevant to trafficking and/ or activation. A little more background re secretory biology could help with this.

>>> As advised by the reviewer we included a statement in the Discussion of the revised manuscript about the pH-values found in the compartments important for PC-trafficking and activation: “After synthesis of the pro-PCs in the endoplasmic reticulum at a pH 7.4 the lower pH of ~6.0 at the late Golgi triggers the auto-activation process ³⁶.” We added the new reference 36 in the revised manuscript.

Minor comments:

-The term PC1 has been replaced in the field by PC1/3 to reflect both the co-discoverers of this enzyme as well as to avoid confusion with the abbreviation for “principal component”.

>>> We have exchanged “PC1” by “PC1/3” throughout the revised manuscript as suggested by reviewer #2.

-Motive and motif are confused throughout the ms.

>>> Was changed in the revised manuscript as suggested by reviewer #2.

-Reference 30 is incorrect; the authors mean to cite Ref 31.

>>> As suggested by reviewer #2 we included Ref. 31 (Rabah, N. et al. 2006) in the Introduction section (line 87 of the original manuscript.). Because Ref. 30 (Levesque, C. et al. 2012) also reports a K_i of the PC1/3-prodomain for furin (Figure 1a in Levesque, C. et al. 2012) we also kept this citation.

- Defining M and F abbreviations in the figure legends would help general readability.

>>> As suggested by reviewer #2 we included an abbreviation definition in the figure legends and tables throughout the revised manuscript. We also write the abbreviations of mutants and fusion proteins consistently in bold letters in the figures and tables of the revised manuscript to avoid any confusion.

Reviewer #3 (Remarks to the Author):

In this manuscript, Klaushofer et al report atomic level structural and functional insights into the basis of proprotein convertase (PC) activation, and report the design and efficacy of a potent new mutant PC1:nanobody fusion protein specific for furin. The results are directly relevant to the challenges of designing specific inhibitors to the broad family of PCs whose dysregulation in vivo is associated with a wide range of diseases. The manuscript is well-written and the results are presented in a logical and descriptive manner.

I feel the manuscript requires the following minor revisions prior to acceptance for publication.

1. Histidine 72:

The authors provide solid X-ray structure and NMR NOE evidence demonstrating the critical role of the side chain of His72 in PC1 which mediates a small hydrogen bond network with nearby residues.

Some questions here:

i. Clearly H72 adopts a $N\epsilon 2H$ neutral tautomer in solution at pH 7.4 based upon the structure, and this must be definitive in the 2D $1H-15N$ HMBC. This is a critical point. Can the authors please add a supplemental figure showing their 2D $1H-15N$ HMBC and the distinctive pattern expected for this tautomer, and also those of the other histidines in this PC1 domain. His75 and His 85 are also nearby in this important secondary cleavage site loop.

>>> We thank the reviewer for this suggestion. We added a figure with the 2D $1H-15N$ HMBC measured at pH 7.4 and 298 K (Figure S11). It shows that His72, His75 and His85 adopt the most common $N\epsilon 2H$ neutral tautomer. We included the statement "A 2D ^{15}N -HMBC experiment revealed the most common neutral $N\epsilon 2H$ tautomeric state for His72, His75 and H75 (Figure S11)." in passage "His72 forms a pH-sensitive hydrogen bond to backbone amide atoms." of the revised manuscript.

ii. The $^{13}C^{15}N$ -labeled PC1 sample affords an excellent opportunity to explore the pK_a 's of the histidines in this domain, via the pH dependence of the $1H/^{13}C$ chemical shift of the side chain $\epsilon 1$'s, which are assigned. Can the authors add this experiment to the Supplemental of this study, given the importance of H72 and the critical role pH plays in the proteolytic processing of PC's.

>>> Following this reviewer's suggestion, we performed a pH titration with the PC1/3-prodomain to determine the pK_a -values of the histidine residues close to the secondary cleavage site loop.

We included this result in the passage “His72 forms a pH-sensitive hydrogen bond to backbone amide atoms.” in the Results section of the revised manuscript: “To investigate the pK_a-values of the histidine residues close to the secondary cleavage site loop of the isolated PC1/3-prodomain we measured 2D ¹⁵N-HMBC spectra of ¹⁵N-labelled wild-type protein in dependence of the pH. Because of the inherent instability and partial unfolding of the protein at room temperature below pH 7.4 (see above), pH-dependent experiments were performed at ~5°C (Figure S12A). Even under optimized conditions we observed unfolding of the protein at pH 6.0, thus the pH-range from 6.3-7.4 was evaluated further. We followed the chemical shifts of the non-protonated nitrogens (Nδ1 in case of the Hε2 tautomer), because they show the most dramatic chemical shift deviations upon protonation, and they report protonation directly and in a predictable manner (>240 ppm if unprotonated and ~180 ppm if positively charged). A fit of the data under the assumption that at pH 2.0 all histidine residues are protonated and that the ¹⁵N chemical shift will be then 176 ppm revealed pK_a-values of 4.82, 5.03 and 5.60 for His72, His75 and His85, respectively (Figure S12B, ³³, ³⁴).”

However, we need to consider that the prodomain is largely influenced by complex formation with furin. This is for instance indicated by a large increase in the melting temperatures and thus in structural stability. It is likely that complex formation influences the pK_a-values of these histidine residues as well. Thus we state in the Discussion section of the revised manuscript: “Interestingly, we observed a relatively low pK_a-value of 4.82 for His72 of the isolated PC1/3-prodomain. This might be different for the furin-bound prodomain due to the influence of the interaction with the catalytic domain on its biochemical properties.”

We also modified the passage “NMR spectroscopy” in the Methods section to describe the pH titration experiments.

2. NMR assignments, especially of the aromatic residues.

i. With the BMRB deposition on hold, please report in the Supplementary your statistics for the percent backbone and overall assignment of the PC1 domain

>>> We included Figure S1 in the revised manuscript illustrating the completeness of the chemical shift assignment.

ii. In addition to the histidines, there are several important NOEs to aromatics reported. How were the aromatic side chains assigned? I am guessing a combination of the Yamazaki et al (1993) experiments (13Cβ -> 1Hδ/Hε) and 13C-NOESY, and the Hδ2's and Hε1's of the histidines assignments fall out of the 2D 1H-15N HMBC. Please describe and cite the methods used.

>>> The aromatic side chains (4×His, 3×His, 3×His, 2×His) were assigned using the very traditional method of a combination of 2D NOESY, 3D ¹³C-edited NOESY, 3D ¹⁵N-edited NOESY (for Trp) 2D COSY and 2D TOCSY spectra (e.g. for His and Trp). Some Cβ resonances were broadened and extremely weak in CBCA(CO)NH and HNCACAB experiments, so that experiments according to Yamazaki might not have worked, at least not for all nuclei. We used only standard experiments and included the additional reference 54 in the Methods section of the revised manuscript.

3. Secondary structure comparison of PC1 from X-ray and NMR (Fig. S8).

i. In the chemical shift indexing (CSI) plot caption please specify for the reader that the rc subscript in the y-axis represents the diamagnetic random coil chemical shift

>>> As suggested by reviewer #3, we added this statement to the legend of Figure S9 in the revised manuscript.

ii. Although +/- 1 residue differences in the exact termini of secondary structural elements are often observed between X-ray and NMR structural data of the same protein, I think the 75-80 region of the

sequence warrants some further experimentation since there is a small strand observed in the Xray structure but “not quite” a strand (gray) in the CSI data. This may reflect some interesting dynamics in this important stretch which is juxtaposed to the critical H72 residue, and contains the secondary cleavage site.

I suggest the authors append residue specific ^{15}N dynamical data in the Fig. S8 plot. i.e., S2 order parameters from ^{15}N T1, T2 and $^{15}\text{N}\{^1\text{H}\}$ heteronuclear NOE data or at the very least the het-NOE.

>>> We agree with the reviewer that the NMR data may point to some population of a strand in the 75-80 region, whereas the crystal structure of the pro-domain at 100 K show a strand. As suggested by the reviewer we included $^1\text{H}\{^{15}\text{N}\}$ heteronuclear NOE data in Figure S9B of the revised manuscript indicating regions with dynamics in the sub-nanosecond timescale. In the Results section of the revised manuscript, we describe this finding: “Whereas the crystal structure reported a β -strand around residues 75-80, the NMR data of that region did not reach the threshold to predict a β -strand (Figure S9A). It could only hint to some population of a β -strand in that region. We measured $^{15}\text{N}\{^1\text{H}\}$ heteronuclear NOE data, a sensitive indicator for dynamics in the sub-nanosecond timescale (Figure S9B). Surprisingly residues 75-80 seem rather rigid (at least in the sub-nanosecond timescale), but Asn74 and His75 show some dynamics.”

However, the binding of the prodomain to the catalytic domain might influence the dynamics of these residues.

4. Linker in the fusion protein:

How did the authors design the linker between PC1 and nb14 in the fusion protein? Is it vulnerable to degradation in vivo and, if so, what can be done to mitigate this without disturbing the efficacy of the fusion protein toward furin inhibition?

>>> We thank reviewer #3 for this question. Due to space limitations, we gave only a minimal description of the linker design in the original manuscript. To attribute this issue in more detail in the revised manuscript: “According to the structures, the N-terminus of the prodomain is only 27 Å away from the C-terminus of the nanobody. In principle, 8 amino acids are sufficient to bridge this distance and link the two proteins. We tested different linker length by energy minimization in COOT. Finally, 11 amino acids were the minimal length to achieve good geometric properties. The linker sequence was chosen to avoid charged amino acids that are prone to degradation by various proteases. On the other hand, amino acids residues were chosen to fit the surface characteristics of the nanobody and of the PC1/3-prodomain at the respective positions. This feature might allow interactions of the linker with the proteins which might reduce its flexibility and thus its accessibility for proteolytic attack.”

Indeed, we also considered the potential vulnerability of this sequence patch during linker design. In the revised manuscript we explain this motivation in the passage describing the linker (see above). In addition, we included an anti-nanobody western blot of the samples shown in Figure 5. Proteolysis in the linker region of the fusion proteins should result in degradation products of ~15 kDa. We report this observation in the Results section of the revised manuscript: “The stable inhibition observed for the fusion proteins indicates that the linker between the prodomain and the nanobody is apparently relatively resistant against proteolytic degradation. This observation is supported by the absence of degradation products in the range of ~15 kDa in the cell culture supernatants (Figure S17).”

In conclusion, the fusion proteins seem to be very stable in cell culture supernatant in the observation period of 72 h. However, this can be different upon internalization into the cell. In the secretory compartments other proteases are present that might digest the fusion proteins in the linker sequence or in other accessible regions. We included a passage in the Discussion section of the revised manuscript: “In the secretory compartments various proteases with broad specificity are present (e.g. the cathepsins,^{38, 39}) that might digest more flexible parts in the fusion proteins (e.g. the linker region or flexible loops).” We included the new references 38 and 39 in the revised manuscript.

5. Minor experimental question: in the main text Methods the authors report using an NOE mixing time of 120ms, but it is 100ms in the Supplemental Information (caption to S13)

>>> We used normally 120 ms, but in this particular case it was 100 ms. In the revised manuscript we write in the Methods section: "... using a mixing time of 120 ms unless stated otherwise."

6. Typos:

i. Fig. 2 caption: Change Asn72 to Asn74 in the text

>>> We changed the numbering of this residue in the figure legend of Figure 2 accordingly.

ii. Labels in Fig. S13 NOESY spectra: please use symbols (Greek letters) where appropriate: i.e. He1 -> Hε1; also, β, γ, δ

>>> The atom nomenclature in Figure S13 (Figure S10 in the revised manuscript) was changed accordingly.

Reviewer #4 (Remarks to the Author):

In this manuscript, Klaushofer et al. structurally investigated the interactions between PC1-25 prodomain and furin, and engineered prodomain-based inhibitors against furin by mutating different amino acid residues near the cleavage site. The activity of the inhibitor was significantly increased by fusion expression with a furin-specific nanobody. Overall, this work deepens our understanding of the mechanism of furin activation and has potential applications.

In terms of inhibiting viral replication, F1 is much better than F2 (Figure 4), and the authors claim that it is largely due to the structural stability of the F1 prodomain [as stated in lines 387-389: "we observed a higher stability of the prodomain part of F1 (M9) in cell culture compared to the prodomain part of F2 (M5)"]; however, a direct comparison of F1 and F2 showed no significant difference in stability in cell culture (Figure 5). Given that the function of furin is performed in the trans-Golgi network (TGN), it is advisable to assess the differences in the intracellular environment: how does F1 reach the site of action? Or what is the half-life of F1 in a cell?

>>> We agree with reviewer #4 that the specific reference to the cell culture experiments in this passage (as cited by the reviewer, lines 387-389 of the original manuscript, Discussion section) is misleading.

In the revised manuscript we changed the sentence cited by the reviewer to specifically emphasize the generally higher structural stability observed for M9 in biochemical assays: "Interestingly, we observed a higher structural stability of M9 (the prodomain part of F1) compared to M5 (the prodomain part of F2) in biochemical assays."

In the revised manuscript we explain in more detail how this can lead to different biological activities:

"Thus, the stability of the prodomain part might be crucial for an antiviral effect of the fusion proteins. This property could be especially important for the half-life of the fusion proteins after cellular uptake. In the secretory compartments various proteases with broad specificity are present (e.g. the cathepsins, ^{38, 39}) which might digest more flexible parts in the fusion proteins (e.g. the linker region or flexible loops). In this context the pH stability and the increased global structural stability of M9 (H72L mutation in combination with (77)A-R-S-A-A secondary cleavage site sequence) are probably important. Without these mutations the pH of ~5.5 as found in endosomes largely destabilizes the PC1/3-prodomain. The prodomain part and especially the secondary cleavage site

loop of F2 (M5) might be partially unfolded and thus inactivated through proteolysis by cathepsins and legumain⁴⁰.”

Furthermore, we completely agree with reviewer #4 that the cellular uptake is probably another important property that could influence the biological activity of the inhibitory proteins. In the revised manuscript we included a statement about the influence of the cellular uptake on the biological activity in the Discussion section: “A more efficient cellular uptake of F1 compared to F2 might also contribute to a higher biological activity.”

However, quantitatively investigating the cellular uptake of these fusion proteins remains challenging, as it would not only require determining their localization in specific cellular compartments but also confirming the functional integrity of the proteins in these compartments. Addressing these questions thoroughly would likely constitute an independent study and thus exceed the scope of the current work. As part of a PhD thesis, we already plan to investigate the cellular availability and localization of these fusion proteins with two complementary approaches: A) Co-localization studies of furin and other PCs with the fusion proteins in the cell by immunofluorescence microscopy and B) co-immunoprecipitation assays of the inhibitory proteins from cellular fractions and western blot analyses. We need to establish many new experiments for this study, and obtaining conclusive data will probably take anywhere from several months to a year or even longer.”

Here are some other minor suggestions for improving the manuscript:
Line 279 “the C-terminus auf the nanobody” should be “the C-terminus of the nanobody”

>>> The typo was corrected accordingly.

Line 315 “with a 10.000-fold reduction” should be “with a 10,000-fold reduction”

>>> The typo was corrected accordingly.

Lines 579-582 ”probably because of huge structural differences to the structure of the human PC1-prodomain of this study ($C\alpha$ -RMSD = 2.9 Å)”: a supplementary figure could be helpful for understanding the “huge structural differences”.

>>> As suggested by the reviewer we included a supplementary figure (Figure S18) in the revised manuscript to show the structural differences of the herein determined structure of isolated the PC1/3-prodomain and of the NMR-structure of the mouse PC1/3-prodomain (PDB-ID 1KN6).

Additional corrections:

>>> We corrected typos throughout the manuscript

>>> In Figure S16A of the revised manuscript (Figure S12A of the original manuscript) we added the unit “nM” to the axis title

>>> Due to a mistake incorrect curve fit lines were exported for Figure S10 in the original manuscript. This was corrected in the corresponding Figure S14 in the revised manuscript. There are only minor differences between the original and the updated figure, which do not affect the significance of the data or the conclusions drawn from them. This issue does not affect the measurement points displayed and their error bars.

>>> A “Data Availability” statement was included in the revised manuscript. The respective statements about PDB deposition and deposition of chemical shift assignments were moved to this section. In addition, we added the statement: “Source data are provided with this paper.”

>>> Because new citations were included in the revised manuscript the numbering of the references was updated throughout the manuscript.

>>> Because new supplementary figures were included in the revised manuscript the numbering of the figures was updated throughout the manuscript.

Reviewer #3 (Remarks to the Author):

“In their revised manuscript, Klaushofer et al made an excellent effort at addressing the questions raised in the first submission, most notably performing a number of additional NMR experiments. The new 2D ^{15}N , ^1H -HMBC experiment in particular is a beautiful result and conclusively reveals the neutral tautomeric state of the important histidines in the key hydrogen bonding network presented in this paper. Although using the $^{15}\text{N}\delta 1$ chemical shift to determine pKa's provides less of a dynamic range than, for example, the $^{13}\text{C}\epsilon 1$ chemical shift, the point is clearly made that the pKa's of H72, H75 and H85 are significantly lower than normal.

One small nomenclature correction to the revisions: for consistency, in line 246 of the revised main text, please change: (...H $\epsilon 2$ tautomer) to (...N $\epsilon 2$ tautomer)”

And the additional provided explanation:

“My nomenclature question is on line 232, p. 8 of the revised MS. I think the two nomenclatures I circled on that page are referring to the same type of histidine neutral tautomer (N $\epsilon 2$ is protonated). I just feel it should be consistent throughout.”

>>> Our response to Reviewer #3:

We thank the reviewer for the positive comments. In the revised manuscript we changed the respective writing to “... (N $\delta 1$ in case of the H $\epsilon 2$ H tautomer)...” to apply consistent atom nomenclature as suggested by the reviewer.